# Measurement of ammonia, amines and iodine species using protonated water cluster chemical ionization mass spectrometry

Joschka Pfeifer[1,2], Mario Simon[2], Martin Heinritzi[2], Felix Piel[2,a], Lena Weitz[2,b], Dongyu Wang[3], Manuel Granzin[2], Tatjana Müller[2], Steffen Bräkling[4], Jasper Kirkby[1,2], Joachim Curtius[2], and Andreas Kürten[2]

[1]CERN, Geneva, 1211, Switzerland
[2]Institute for Atmospheric and Environmental Sciences, Frankfurt Am Main, 60438, Germany
[3]Paul Scherrer Institute, Villigen, 5232, Switzerland
[4]TOFWERK AG, Thun, 3600, Switzerland

[a]now at: Ionicon Analytik GmbH, 6020 Innsbruck, Austria
[b]now at: GSI Helmholtzzentrum für Schwerionenforschung GmbH, Darmstadt, 64291, Germany

*Correspondence to*: Joschka Pfeifer (joschka.pfeifer@cern.ch)

**Abstract.** Here we describe the design and performance of a new water cluster Chemical Ionization-Atmospheric Pressure interface-Time Of Flight mass spectrometer (CI-APi-TOF). The instrument selectively measures trace gases with high proton affinity such as ammonia and dimethylamine, which are important for atmospheric new particle formation and growth. Following the instrument description and characterization, we demonstrate successful measurements at the CLOUD (Cosmics Leaving OUtdoor Droplets) chamber where very low ammonia background levels of ~4 pptv were achieved (at 278 K and 80% RH). The limit of detection of the water cluster CI-APi-TOF is estimated to be ~0.5 pptv for ammonia. Although no direct calibration was performed for dimethylamine (DMA), we estimate its detection limit is at least 3 times lower. Due to a short ion-molecule reaction time (< 1ms) and high reagent ion concentrations ammonia mixing ratios of at least up to 10 ppbv can be measured without significant reagent ion depletion. Besides the possibility to measure compounds like ammonia and amines (dimethylamine), we demonstrate that the ionization scheme is also suitable for the measurement of trace gases containing iodine. During CLOUD experiments to investigate the formation of new particles from $I_2$, many different iodine-containing species were identified with the water cluster CI-APi-TOF. The compounds include iodic acid as well as neutral molecular clusters containing up to four iodine atoms. However, the exact molecular composition of the iodine-containing clusters are ambiguous, due to the presence of an unknown number of water molecules. The quantification of iodic acid ($HIO_3$) mixing ratios is performed from an inter-comparison with a nitrate CI-APi-TOF. Using this method the detection limit for $HIO_3$ can be estimated as 0.007 pptv. In addition to presenting our measurements obtained at the CLOUD chamber, we discuss the applicability of the water cluster CI-APi-TOF for atmospheric measurements.

## 1 Introduction

Ammonia ($NH_3$) is an important atmospheric trace gas that is mainly emitted by agricultural activity due to animal husbandry and the use of fertilizers, and by vehicles in urban environments. It can partition to the aerosol phase and is one of the most important compounds contributing to secondary aerosol formation (Jimenez et al., 2009). Strong reductions in $PM_{2.5}$ and the associated adverse health effects could potentially be achieved by decreasing ammonia emissions (Pozzer et al., 2017). However, ammonia is not only partitioning to existing particles, but is also a key vapor driving new particle formation due to its stabilization of newly-formed clusters in ternary (sulfuric acid-water-ammonia) and multi-component (sulfuric acid-water-ammonia-highly oxygenated organic molecules) systems (Kirkby et al., 2011; Kürten et al., 2016a; Lehtipalo et al., 2018). On a global scale, a large fraction of newly formed particles and cloud condensation nuclei involves ammonia (Dunne et al., 2016). The involvement of ammonia in nucleation has recently been measured in the free troposphere, in Antarctica, and in the boreal forest (Bianchi et al., 2016; Jokinen et al., 2018; Yan et al., 2018). In the upper troposphere, model calculations suggest that ammonia is important for new particle formation and early growth (Dunne et al., 2016). Recent satellite measurements support this finding by the observation of up to several tens of pptv (parts per trillion by volume) of ammonia over Asia (Höpfner et al., 2016). Ammonia has a very strong effect on nucleation involving sulfuric acid and water, e. g., recent studies have shown that very low amounts of $NH_3$ in the pptv-range, or even below, can enhance nucleation rates by orders of magnitude compared with the pure binary system of sulfuric acid and water (Kirkby et al., 2011; Kürten et al., 2016a; Kürten, 2019). Stronger basic compounds like amines or diamines, have been shown to enhance nucleation rates even stronger compared to $NH_3$, despite their much lower atmospheric concentrations (Almeida et al., 2013; Kürten et al., 2014; Jen et al., 2016; Yao et al., 2016). The experimental measurements are confirmed by quantum chemical calculations that compare the stabilizing effects of ammonia, amines, and diamines (Kurtén et al., 2008; Elm et al., 2017; Yu et al., 2018). For these reasons the measurement of these compounds is required in order to understand new particle formation and the partitioning between the gas and aerosol phase. It is important to note that ammonia can easily exceed several ppbv in the boundary layer, whereas amine mixing ratios are typically present at a few pptv only (Ge et al., 2011; Hanson et al., 2011; You et al., 2014; Kürten et al., 2016b; Yao et al., 2016).

In some previous studies, ammonia has been measured using optical absorption or chromatographic methods (Norman et al., 2009; Bobrutzki et al., 2010; Verriele et al., 2012; Bianchi et al., 2012; Pollack et al., 2019). These measurement techniques are often specialized for the detection of only a few selected compounds, whereas chemical ionization mass spectrometry (CIMS) can often measure a suite of atmospheric trace gases simultaneously at low concentrations and high time resolution. The use of different reagent ions has been described in the literature for ammonia and amine measurements, e. g., protonated acetone, protonated ethanol, $O_2^+$, and protonated water clusters have been successfully applied (Nowak et al., 2007; Norman et al., 2007; Benson et al., 2010; Hanson et al., 2011; You et al., 2014; Yao et al., 2016). Nowak et al. (2010) deployed their instrument on an aircraft for measurements at up to ~5 km altitude. The limit of detection (LOD) varies between 35 pptv (You et al., 2014) and 270 pptv (Norman et al., 2009) for ammonia, whereas dimethylamine (and other amines) can be detected in

the sub-pptv range (You et al., 2014; Sipilä et al., 2015; Simon et al., 2016). In this study we introduce a newly developed chemical ionization mass spectrometer that uses protonated water clusters for selective ionization of ammonia and dimethylamine. The instrument is based on a high resolution Chemical Ionization-Atmospheric Pressure interface-Time Of Flight mass spectrometer (CI-APi-TOF, Aerodyne Inc. and TOFWERK AG) combined with a home-built ion source. The instrument is termed water cluster CI-APi-TOF, in accordance with other established techniques using the same mass

spectrometer but different reagent ions, e. g., the nitrate CI-APi-TOF for sulfuric acid, highly-oxygenated organic molecule, and cluster measurements (Jokinen et al., 2012; Ehn et al., 2014; Kürten et al., 2014).  Here we describe and characterize the instrument during experiments performed at the CLOUD (Cosmics Leaving OUtdoor Droplets) chamber at CERN (European Organization for Nuclear Research). We show that the detection limit for ammonia is below 1 pptv, which is unprecedented to our knowledge. Besides the measurement of basic compounds with high proton affinity, it was observed that the protonated

water clusters are also well-suited to measure iodine-containing species such as iodic acid ($HIO_3$) and neutral molecular clusters containing up to four iodine atoms. The corresponding signals in the mass spectra were identified during CLOUD experiments on new particle formation from the oxidation of molecular iodine. The relevance of such compounds for nucleation in the atmosphere has recently been reported (Sipilä et al., 2016). Our findings indicate that the water cluster CI-APi-TOF can provide sensitive real-time measurements of several trace gases that are important for atmospheric new particle formation and growth:

ammonia, amines (dimethylamine) and iodine species.

## 2. Methods

### 2.1 CLOUD chamber

    The measurements presented here were carried out at the CLOUD (Cosmics Leaving OUtdoor Droplets) chamber at CERN

(European Organization for Nuclear Research) during fall 2017 (CLOUD12 campaign) and fall 2018 (CLOUD13 campaign). The CLOUD chamber is used to investigate new particle formation from different trace gas mixtures under controlled atmospheric conditions regarding temperature, relative humidity, UV light intensity and ionization level (Kirkby et al., 2011; Kupc et al., 2011, Duplissy et al., 2016). The cylindrical stainless steel chamber has a volume of 26.1 $m^3$. It is designed to ensure that trace gas contaminants are low enough to allow precisely controlled nucleation experiments (Kirkby et al., 2016).

The chamber is continuously flushed with synthetic air generated from liquid nitrogen and oxygen. The temperature and relative humidity of the air inside the chamber can be precisely controlled. For the present study, ammonia and dimethylamine from gas bottles were injected using a two-step dilution system (Simon et al., 2016; Kürten et al., 2016a). The calibration of the water cluster CI-APi-TOF with ammonia (Figure 1) was carried out while the instrument was disconnected from the chamber. For the calibration measurements, the two-step dilution system from the CLOUD chamber was replicated (Figure 1

and Section 2.2).

Iodine is introduced into the chamber by nitrogen flowing over solid, molecular iodine ($I_2$, Sigma-Aldrich, 99.999% purity) placed in a stainless steel evaporator immersed in a water bath at 303 K, with a temperature stability near 0.01 K. The generation of iodine-containing species for new particle formation is initiated by photolysis of $I_2$ in the presence of ozone and water. Measurements presented here were carried out at chamber temperatures between 223 and 298 K, with relative humidity (RH) ranging between 10 and 90%. A Pt100 sensor string measured the air temperature in the CLOUD chamber (Dias et al., 2017).

A chilled dew point mirror (Edgetech Instruments) measured the dew point inside the CLOUD chamber. The relative humidity is derived from water vapor pressure formulations published by Murphy and Koop (2005). Additionally, the RH was measured by a Tunable Diode Laser system (TDL) developed by Karlsruhe Institute for Technology (KIT), which was installed in the mid plane of the chamber (Skrotzki, 2012). The relative humidity was derived using the mean value of both instruments, with a combined measurement uncertainty of 5%.

## 2.2 Water cluster CI-APi-TOF

The selective detection of ammonia and amines by atmospheric pressure chemical ionization using positively-charged water clusters has been demonstrated recently (Hanson et al., 2011). The same ionization technique is used in the present study. The reagent and product ions are measured with an Atmospheric Pressure interface-Time Of Flight mass spectrometer (APi-TOF), which is coupled with a newly-designed crossflow chemical ionization (CI) source operated at ambient pressure (Figure 1). The reagent ions, i.e., protonated water clusters (($H_2O)_nH_3O^+$) are generated by positive corona discharge in the presence of argon (95 %), oxygen (5 %), and water vapor. The water vapor is added by bubbling the argon through a stainless steel humidifier (containing about 1 liter of Millipore purified water) held at ambient temperature of ~20 °C. As suggested by Hanson et al. (2011), a few droplets of sulfuric acid were added to the water in order to minimize potential contamination with ammonia from the water supply. Flow rates of 2.5 standard liters per minute (slm) for argon and 0.1 slm for oxygen were used, respectively. All flow rates were controlled by calibrated mass flow controllers (MFC). A conversion factor for the measured argon flow (provided by the MFC manufacturer) was applied. First attempts have been made using nitrogen instead of argon for the flow that passes the corona needle, but this resulted in much higher ammonia backgrounds. These backgrounds are most likely explained by $NH_3$ production involving $N_2$ in the corona plasma. Furthermore, the addition of oxygen is necessary for the generation of a stable corona discharge in positive mode when using argon as the main ion source gas (Weissler, 1943).

Protonated water is also used in proton-transfer-reaction mass spectrometry (PTR-MS) that has been described in numerous publications (Good et al., 1970; Kebarle, 1972; Zhao and Zhang, 2004; Hansel et al., 2018). A simplified reaction scheme leading to the formation of protonated water clusters is shown below (Sunner et al., 1988):

$$e^- + O_2 \rightarrow O_2^+ + 2\,e^-, \tag{R1}$$

$$O_2^+ + (H_2O)_2 \rightarrow O_2^+(H_2O)_2 \rightarrow H_3O^+(OH) + O_2, \tag{R2}$$

$$H_3O^+(OH) + (H_2O)_n \rightarrow H_3O^+(H_2O)_n + OH. \qquad\qquad (R3)$$

The PTR-MS operates its ion-molecule reaction zone (IMR) typically at low pressure (~ 10 hPa) and uses an electric field (~ 100 V mm$^{-1}$) to break up water clusters such that mainly $H_3O^+$ ions react. The use of charged water clusters (($H_2O)_{n\geq1}H_3O^+$ instead of $H_3O^+$) can increase the selectivity as water clusters have a much higher proton affinity compared with the water

monomer (Aljawhary et al., 2013). However, due to their high proton affinity, ammonia and amines can still be detected according to the following reaction scheme:

$$H_3O^+(H_2O)_n + X \rightarrow XH^+(H_2O)_b + (n + 1 - b) \cdot (H_2O), \qquad\qquad (R4)$$

where X represents the substance that is ionized in the ion-molecule reaction zone (see below) and detected in the mass spectrometer. As water molecules can evaporate in the atmospheric pressure interface of the mass spectrometer, some of the product ions can be detected without water, e. g., ammonia is mainly detected as $NH_4^+$ (see Figure 2).

A schematic drawing of the calibration setup and the ion source is shown in Figure 1. The gas mixture for the ion source is composed of argon, oxygen and water vapor. It is introduced from two lines placed in the opposite direction to each other

at an overall flow rate of ~2.6 slm (Figure 1a). The Electrodes of the ion source are displayed in red colors in Figure 1b. The connection to the mass spectrometer is shown using blue color. The 1" sampling line and the inlet (22 mm inner diameter) consist of stainless steel and are shown in green color. Components used for insulation are shown in white colors. A total sample flow rate of ~ 19.5 slm is maintained by a vacuum pump and a mass flow controller. The overall length of the sampling line connecting the CLOUD chamber and the ion molecule reaction zone is 1.3 m. A voltage of 3600 V is applied to the corona

needle while 500 V are applied to the conically-shaped counter electrode (Electrode 1 in Figure 1b) made of stainless steel. The housing of the ion source is made of polyether ether ketone (PEEK). The ion source gas and the generated reagent ions flow through a funnel (smallest inner diameter 2.5 mm) before they mix with the sample flow. A small capillary (inner diameter of 0.8 mm) is located opposite of the funnel (Electrode 2 in Figure 1b). The electric field between the counter electrode and the capillary (at ground potential) accelerates the ions towards the entrance of the mass spectrometer. The pinhole plate (pinhole

inner diameter of 350 µm) and the capillary are in electric contact and ~0.8 slm flow through the capillary and the pinhole into the mass spectrometer. The measured product ions are generated in the ion-molecule reaction zone (IMR, yellow area in Figure 1a) at atmospheric pressure. The dimension of the IMR is defined by the distance between the counter electrode and the capillary (~ 16.4 mm). After passing the pinhole, the ions are transported through two quadrupoles (Small Segmented Quadrupole, SSQ and Big Segmented Quadrupole, BSQ) towards the detection region of the mass spectrometer (Micro-

Channel Plate, MCP; pressure is approx. $1\times10^{-6}$ hPa). The estimated reaction time is <1 ms. This short reaction time allows the measurement of high ammonia mixing ratios (up to ~10 ppbv) without significant depletion of the reagent ions, which would be the case when using an ion source design for the measurement of sulfuric acid (Eisele and Tanner, 1993; Kürten et

al., 2011), which is typically present at much lower concentrations than ammonia. The principle of a cross-flow ion source was introduced by Eisele and Hanson (2000) who used this technique to detect molecular sulfuric acid clusters. In more recent

studies, this technique was used for the measurement of ammonia (Nowak et al., 2002; Nowak et al., 2006; Hanson et al., 2011).

The measured volume mixing ratio (VMR, in pptv) of detected compounds is derived from a calibration factor ($C$) and the sum of the product ion counts per second (pcs) normalized against the sum of the reagent ion counts per second (rcs) (Kürten et al., 2016b; Simon et al., 2016):

$$VMR = C \cdot \ln\left(1 + \frac{\sum pcs}{\sum rcs}\right) = C \cdot ncps. \tag{1}$$

Equation (1) yields the VMR measured by the water cluster CI-APi-TOF as a function of the normalized counts per second ($ncps$). A calibration factor, $C$, which includes factors like the reaction rate and the effective reaction time, is required to convert the $ncps$ to a mixing ratio. This factor can be derived from the inverse slope of a calibration curve (see Section 3.2). While Hanson et al. (2011) report a maximum for the water cluster distribution at the pentamer, evaporation of water seems

to be stronger in our instrument. The maximum signal in clean spectra is usually found for the water dimer (($H_2O$)$H_3O^+$, see Figure 2) and a strong drop in the reagent ion signals is found beyond the tetramer (($H_2O$)$_3H_3O^+$). Therefore, the sample quantification includes, using ammonia as an example, the product ions ($H_2O$)$_nNH_4^+$ with $n = 0,1$ and the reagent ions ($H_2O$)$_mH_3O^+$ with $m = 0\text{-}3$. Possible losses in the sampling line are not taken into account by the calibration factor (see Section 3.8 for discussion of sampling line losses). The only compound for which a direct calibration is performed in the present study

is ammonia (Section 2.3). When mixing ratios for dimethylamine are presented, the same calibration factor is used. This approach can introduce uncertainty as the proton affinity, as well as the transmission efficiency differ for dimethylamine compared to ammonia. However, previous studies showed that the ionization efficiency from protonated water clusters is collision-limited for both ammonia and dimethylamine (Sunner et al., 1988; Hanson et al., 2011). The applicability of this approach is discussed in Section 3.9; it is estimated that the mixing ratios for dimethylamine are correct within a factor of ~3.5.


## 2.3 Calibrations

### 2.3.1 Ammonia

Figure 1a shows a schematic drawing of the experimental setup during the calibrations with ammonia. The ammonia was taken from a gas bottle containing an $NH_3$ mixing ratio, $B$, of 100 ppmv diluted in pure nitrogen (Air Liquide, ±5% uncertainty for

the certified $NH_3$ mixing ratio) that is diluted in two steps, where MFCs (indicated as $M_n$ in Figure 1a) are used to obtain different set points for the volume mixing ratio (Figure 1a). During the second dilution step the mixture from the first dilution is injected into the center of the main sample flow (flow rate, $Q_{sample}$). The theoretical VMR$_{theor}$ is given by (Simon et al., 2016):

$$VMR_{theor} = \frac{M_1}{M_1+M_2} \cdot \frac{M_3}{M_3+Q_{sample}} \cdot B. \tag{2}$$


The flow of ammonia from the gas bottle is adjusted by $M_1$ (0.01 slm max.), whereas $M_2$ (2 slm range) controls the flow of nitrogen for the first dilution step. The flow of diluted ammonia that is introduced into the sample flow is controlled by $M_3$ (0.1 slm range). The calibration flow consists of the same synthetic air that is used for the CLOUD chamber. The flow is provided by two MFCs that control a dry portion ($M_4$) and a wet portion of the flow that has passed a stainless steel water

bubbler ($M_5$, see Figure 1). By adjusting $M_4$ and $M_5$ (both 50 slm range) the RH of the sample flow can be controlled in order to test whether a humidity dependence exists for reaction (R4). Care is taken that the sum of $M_4$ and $M_5$ is always somewhat larger than $Q_{sample}$. To avoid overpressure in the sampling line, the excess flow is vented through an exhaust before the sampling line.

Accordingly, the measured sample air consists of synthetic air (80% nitrogen, 20% oxygen) with adjustable RH and

ammonia mixing ratio. To obtain calibration curves, the highest targeted ammonia mixing ratio is adjusted first. The calibration points are then recorded by stepping down the flow of $M_3$. In this way, equilibration times are fairly short as the $NH_3$ mixing ratio before and directly after $M_3$ remains constant (see also Section 3.2).

### 2.3.2 Iodine Oxides

Besides the detection of ammonia and dimethylamine by the described ionization scheme, product ions from iodine-

containing species were detected during new particle formation experiments initiated from $I_2$ photolysis during the CLOUD13 run. Prominent signals ($HIO_3 \cdot H^+$ and $HIO_3 \cdot H_3O^+$) corresponding to the neutral species of $HIO_3$ were observed among others (discussed in Section 3.6 and Table 2). These species can be unambiguously identified due to the large negative mass defect of the iodine atom and the high mass resolution (> 3000 Th/Th). No direct calibration for $HIO_3$ was performed; however, another chemical ionization mass spectrometer using nitrate reagent ions (nitrate CI-APi-TOF) was also measuring $HIO_3$ at

CLOUD. Therefore, a calibration factor for $HIO_3$ is derived by scaling concentrations measured by the nitrate CI-APi-TOF that is calibrated for sulfuric acid (Kürten et al., 2012). We further assume that both sulfuric and iodic acid are detected with the same efficiency by the nitrate CI-APi-TOF. This assumption introduces uncertainty when estimating the detection limit of $HIO_3$. However, as the reaction of sulfuric acid with nitrate ions is at the kinetic limit, the detection limits shown here based on this assumption can be seen as lower limits. Unfortunately, there is currently no direct calibration technique established for

iodic acid in the gas phase. Thus, the same assumption as in a previous study for deriving gas phase concentrations of iodic acid are used in the present study (Sipilä et al., 2016).

For the instrument inter-comparison (and the indirect calibration of the water cluster CI-APi-TOF), 18 different CLOUD experimental runs were chosen and mean values were calculated for different steady-state concentrations. We took 6 steady-state concentrations each at temperatures of 263 K (80% RH) and 283 K (40% RH and 80% RH).

**2.4 PICARRO**

A PICARRO G1103-t $NH_3$ Analyzer (PICARRO Inc., USA) measured ammonia mixing ratios based on cavity-ring down spectroscopy during CLOUD12 and CLOUD13. The instrument is suitable for real-time monitoring of ammonia in ambient air and has been presented in previous studies (Bell et al., 2009). The G1103-t was installed at the CLOUD chamber with its own sampling line coated with Sulfinert (Restek GmbH, Germany), where the coating reduced the losses of ammonia to the

sampling line walls considerably. Since the PICARRO has a rather small sample flow rate (< 1 slm), an additional pump was used to enhance the flow rate to 5 slm just before the instrument inlet. This was done in order to minimize line losses and to decrease the response times. It was, however, not quantitatively tested in how far these measures (Sulfinert and increased flow rate) helped with the measurements. The PICARRO was also independently calibrated with a $NH_3$ permeation tube (Fine Metrology, Italy) using a multigas calibrator (SONIMIX 6000 C1, LNI Swissgas, Switzerland). The time interval for one

measurement of the PICARRO is 5 seconds for which a lower detection limit of 200 pptv is reported (PICARRO Inc., USA; Martin et al., 2016). By using the same method (at the same time period) as shown for the water cluster CI-APi-TOF (see Section 3.5), we derive a detection limit of 366.2 pptv for the PICARRO unit used in this study.

## 3. Results and Discussion

### 3.1 Main peaks in spectrum

Figure 2 shows a typical spectrum during calibrations, where 10 ppbv of ammonia are injected (40% RH, ~293 K). The most prominent primary ions are $H_3O^+$, $(H_2O)H_3O^+$ and $(H_2O)_2H_3O^+$. $(H_2O)H_3O^+$ is the dominant primary ion in the mass spectrum. The water tetramer ($(H_2O)_3H_3O^+$) is usually the largest water cluster that can be detected. The addition of ammonia generates $NH_4^+$ and $(H_2O)NH_4^+$; a small signal from $NH_3^+$ is also visible. At low ammonia concentrations the signal from $NH_4^+$ can have a similar magnitude compared with the signal from $H_2O^+$ (possibly from reactions of $O_2^+$ and $H_2O$). Since these ions

have the same integer mass, a high mass resolving power is essential in terms of reaching low detection limits as otherwise the differentiation between the two signals is not possible. At the low masses the APi-TOF used in the present study reaches a resolving power of ~2000 Th/Th, which is sufficient to separate the two peaks. For the analysis of the spectra, the software TOFWARE is used that allows analyzing high resolution spectra (Stark et al., 2015; Cubison and Jimenez, 2015; Timonen et al., 2016). Prominent peaks from $N_2H^+$, $NO^+$ and $O_2^+$ can also be found in the spectrum shown in Figure 2. It is not clear how

these ions are formed and why they survive the relatively long reaction time of ~1 ms since Good et al. (1970) report that $O_2^+$ reacts rapidly away in moist air. For the analysis the presence of these background peaks is currently ignored and they are not counted as reagent ions (in equation (1)) as no evidence exists that they interact with the target species relevant for the present study. An exception could be $NH_3^+$ (possibly from reaction of $O_2^+$ and $NH_3$); but $NH_3^+$ is not considered and is only a small fraction of $NH_4^+$.

In contrast to the spectrum shown in Figure 2 with relatively small water clusters, Hanson et al. (2011) observe the highest signal in the water cluster distribution for the pentamer. We explain this difference in more pronounced fragmentation and evaporation of ion clusters in the atmospheric pressure interface of our mass spectrometer.

For estimating an ammonia mixing ratio according to equation (1), the product ion count rates are normalized against the dominating reagent ion count rates. Figure 2 shows that even at 10 ppbv of injected ammonia the reagent ion signals are

significantly higher than the product ion count rates. This indicates that no significant reagent ion depletion occurs and thus the normalized counts per second respond linearly with the ammonia VMR at least up to 10 ppbv (see Section 3.2).

## 3.2 Ammonia and iodic acid calibration

Figure 3 shows the calibration curves obtained for $NH_3$ and $HIO_3$ (for the CLOUD13 campaign). Each dot represents the mean value of a steady state measurement of at least 20 minutes. The normalized counts per second are based on the two

highest signals assigned to the analyzed compound (ammonia: $NH_4^+$ and $(H_2O)NH_4^+$; iodic acid: $HIO_3H^+$ and $HIO_3H_3O^+$). The total error of the VMR (on the $x$-axis) is calculated by Gaussian error propagation taking into account the standard deviation of the flow rates from the mass flow controllers and the uncertainty of the VMR inside the ammonia gas bottle. Since we obtained the VMR shown in Figure 3b by scaling the concentrations measured by a nitrate CI-APi-TOF calibrated for sulfuric acid, the error on the $x$-axis equals the uncertainty of these measurements (estimated as a factor of two for the iodic

acid concentration). The error on the $y$-axis is given by the standard deviation of the normalized counts per second. We derive a calibration curve from a linear regression model using the Wilkinson-Rogers Notation (Wilkinson and Rogers, 1973). The fit is forced through the origin; however, even when the fit is not constrained, the resulting slope is essentially the same (the results for the slopes/sensitivities differ by 1.35%). The derived slopes represent the inverse of the calibration factor, $C$ (~ $1.46\times10^5$ pptv at 40% RH), in equation (2). Figure 3 shows that all measured mixing ratios lie in the area of the confidence

intervals (95% confidence intervals) and thus the linear model describes the dependency very well. The calibration was performed before the CLOUD13 experiment (Sept. 2018), during and after the experiment (Dec. 2018) at different levels of humidity (calculated relative humidity levels between ~3% and 82%) and ambient temperatures. The calibrations for ammonia were performed by introducing the highest mixing ratio first. However, it took almost a day to reach stable signals as the tubing and the two MFCs through which the ammonia was flown ($M_1$ and $M_3$) needed to equilibrate. The further calibration points

were then recorded by reducing the flow rate of $M_3$. In this way, no change in the ammonia mixing ratio inside the capillary before the main sampling line and in the MFCs was necessary. This allowed for a relatively fast stepping through the calibration set points. However, even when the ammonia flow was shut-off there was still significant diffusion of ammonia from the capillary into the sampling line, which resulted in relatively high background values (with nominally zero $NH_3$). For this reason, we derived the limit of detection by measuring the background of the CLOUD chamber with the calibration lines

removed (Section 3.5). During the calibrations, the relative humidity was calculated by assuming that the sample flow passing the water reservoir is saturated with water (Figure 1a). For the calibrations carried out after the campaign, the temperature of the total sample flow was measured to derive the absolute humidity. The calibration points in Figure 3 were taken at measured

gas flow temperatures of 288 to 290 K. The relative humidity was set to 40% RH by adjusting the dry and the wet flow rates for the sample flow; these conditions correspond to an absolute humidity of ~ 0.0057 kg m$^{-3}$. The calibration factor derived for CLOUD12 (for ammonia) differs from the calibration factor shown here. This is due to a different ion source (designed for a 0.5'' sampling line in CLOUD12 compared with a 1'' line in CLOUD13), a different sample flow rate and different tuning of the CI-APi-TOF.

### 3.3 Response times

The response time of the water cluster CI-APi-TOF is defined as the characteristic time needed for the instrument to react on changes in the ammonia mixing ratio. The response time takes into account two processes. These are the time needed until the instrument reacts on changes in the mixing ratio and the time needed until a steady state is established in the lines. In the following, we define the response time as the time required for the instrument to reach 95% of the new mixing ratio being injected. Figure 4 indicates the typical response times of the water cluster CI-APi-TOF during calibrations (here at 60% relative humidity). It shows a decay between two calibration steps when the injected ammonia is reduced from 9509 pptv to 6911 pptv and a rise in the signal when the ammonia mixing ratio is increased from 500 pptv to 9509 pptv. Panel a) indicates a clear difference between the time needed until the instrument reacts on the changes in the mixing ratio (red line) and the time needed until the lines reach 95% of the new steady state (black line). We expect the same behavior for a decay from 9509 to 500 pptv, however, the mixing ratios were gradually reduced during calibrations. Thus, for the gradual decays, the time needed for the lines to reach a new equilibrium is rather short. While the variation of instrumental response time is small (6 to 10 seconds for decays from 9509 to 6911 pptv and 18 to 25 seconds for a rise from 500 pptv to 9509 pptv, respectively), the time until a steady state is established in the lines varies depending on precursor conditions and relative humidity (see Section 3.8). Thus, an estimation of a response time can vary significantly. In our experiments, the response times (including both processes described above) during a rise in ammonia mixing ratio varied between 535 seconds (20% relative humidity) and 890 seconds (60% relative humidity, shown in Figure 4). For a decay of ammonia mixing ratio from 9509 to 6911 pptv the response times vary between 37 seconds and 54 seconds.

### 3.4 Influence of the humidity on the sensitivity

Figure 5 shows the sensitivity dependence for the ammonia measurements with varying relative humidity. These data points are derived from calibration curves similar to the one shown in Figure 3a. However, during the calibrations the humidity was changed by adjusting the dry and wet sample flow rates. For all conditions $NH_4^+$ has the highest product ion count rate. However, the ratio of the signals for $(H_2O)NH_4^+$ and $NH_4^+$ increases with humidity as well as the sensitivity. A possible explanation for the observed sensitivity dependence could be increased collision rates at high humidity where larger water clusters are present. In addition, the detection efficiency as function of the ion mass can vary depending on the voltages applied to the ion source and the APi-section, as well as the time of flight region of the mass spectrometer. Thus, the mass spectrometer

does not have a constant detection efficiency over the full mass range (Heinritzi et al., 2016). A higher detection efficiency at $m/z$ 36 $((H_2O)NH_4^+)$ compared with $m/z$ 18 $(NH_4^+)$ together with the higher fraction of $(H_2O)NH_4^+$ compared with $NH_4^+$ at high humidity could explain some of the observed effect. However, the observed increase in sensitivity is not dramatic (increase by a factor of ~2.5 when the humidity increases by a factor of 10). Nevertheless, the effect is taken into account by using the measured relative humidity inside the CLOUD chamber (see Section 2.1) to correct the derived ammonia mixing ratio. The effect of temperature on the sensitivity could not be tested during a dedicated calibration experiment as our calibration setup is not temperature-controlled. However, during a transition from high to low temperature in the CLOUD chamber and constant ammonia injection, no significant change in the measured ammonia was observed, which indicates a weak influence of temperature.

The dependency of the sensitivity with relative humidity and temperature is different for the measurement of iodic acid as shown in Figure 6. While $NH_4^+$, without a water molecule, is the dominant signal for ammonia, $H_4IO_4^+$, which is $H_2O \cdot HIO_3H^+$ or $HIO_3 \cdot H_3O^+$, yields the highest signal for iodic acid. We observed an increasing sensitivity at lower temperatures, while the humidity dependency appears to be smaller compared to the ammonia measurements (Figure 5). The higher count rate of $H_4IO_4^+$ compared to $H_2IO_3^+$ could indicate that iodic acid requires additional water in order to be associated with a positively charged ion. However, during the transition from ambient pressure into the vacuum of the mass spectrometer, water molecules can evaporate and leave $H_2IO_3^+$ in a non-equilibrium process. Besides the observation of iodic acid, additional signals from iodine-containing species can be found in the spectra. These species are listed in Table 2. Elucidating the exact formation pathways of these ions and the corresponding neutral species is subject for future work.

## 3.5 Detection limits and instrumental backgrounds

Determining the limit of detection (LOD) for ammonia is not trivial as the background signal is not constant. During the calibrations a relatively high background was observed, which was usually decreasing slowly after the ammonia flow through the capillary was shut off. A typical value reached a couple of minutes after the ammonia flow was turned off is ~30-60 pptv. When the water cluster CI-APi-TOF was connected to the CLOUD chamber the NH$_3$ signals usually dropped significantly when no ammonia was actively added. However, even under these conditions the ammonia was not zero and the measured signal changed when the RH or temperature of the chamber was adjusted. There is evidence that the contaminant level of the CLOUD chamber with respect to ammonia is on the order of several pptv at 278 K and 38% RH (Kürten et al., 2016a). During CLOUD13 the measured background (at 278 K and 80% RH) was 3.7 pptv, which in principle confirms the previous estimates. The fact that the sampling line of the instrument can also be a source of contamination could explain the somewhat higher value. Another source of background ammonia could be the ion source. During the early stages of our development we used nitrogen instead of argon as the main ion source gas. This led to ammonia backgrounds of several 100 pptv since ammonia can be generated by the nitrogen plasma from the corona tip when it mixes with the humid sample flow (Haruyama et al., 2016). However, replacing nitrogen with argon quite drastically decreased the background ammonia signals. Still, traces of nitrogen containing gases in the ion source could contribute to the ammonia background. For the present study we report a

background ammonia mixing ratio of 3.7 pptv (Table 1) for 278 K and 80% RH but note that the background is significantly lower for lower chamber temperatures, which argues against the ion source being a significant source of ammonia since it is always at ambient temperature.

The LOD is defined as the additional ammonia mixing ratio that is necessary to exceed three times the standard deviation at background conditions (You et al., 2014). This value corresponds to 0.5 pptv for an averaging time of 1 minute. Assuming the same sensitivity as for ammonia and taking into account the background signals for the exact masses, LODs for other compounds can be estimated. Besides the calculated values for ammonia, Table 1 lists the estimated backgrounds and LODs for dimethylamine and iodic acid. The evaluation of high resolution data is necessary for deriving the values shown in Table 1 as for some compounds several species occur at the same integer mass. For example, for dimethylamine (exact mass of protonated compounds at 46.0651 Th) other species like $NO_2^+$ (45.9924 Th) or $CH_4NO^+$ (46.0287 Th) can interfere. For dimethylamine only the peak with the highest count rate is taken into account, since $(C_2H_7N)H_3O^+$ interferes with other compounds measured during the experiments even when using high resolution data. In principle, the omission of the larger product ions (with one additional water molecule) should lead to a different calibration constant. However, the effect is small as the ion signals with the associated water are smaller than the products without the water molecule for the measured bases. The applicability of the assumption (using the calibration constant derived for ammonia for dimethylamine) is discussed in Section 3.9.

The instrumental background for $NH_3$ is higher than the estimated backgrounds for the other compounds shown in Table 1. This might be the case, since low levels of $NH_3$ are more difficult to achieve due to the ubiquitous presence of ammonia. In any case, the detection limit derived for ammonia is well below the LOD reported for other measurement techniques and instruments (Bobrutzki et al., 2010; You et al., 2014; Lin; Wang et al., 2015). However, the performance of the water cluster CI-APi-TOF during atmospheric measurements remains to be tested. The signal of dimethylamine is most of the time below the estimated limit of detection.

The estimated detection limit of iodic acid is well below the LOD calculated for ammonia and dimethylamine (Table 1). We might explain this when looking at signals that could possibly interfere with the measured compounds. All compounds shown in Table 1 have an integer mass, where other signals are also detected, e.g., $H_2O^+$ at the nominal mass of ammonia, or $NO_2^+$ at the nominal mass of dimethylamine. For the high masses of the iodine containing species with their strong negative mass defects these isobaric compounds are much less crucial. Additionally, iodic acid has a much lower vapor pressure compared with ammonia and is not emitted efficiently from surfaces at temperatures relevant for the present study. Therefore, much lower backgrounds can be expected even if the sampling line and the instrument were exposed to high concentrations before.

**3.6 Identified iodine species during CLOUD13**

The signals for $HIO_3$ measured by the water cluster CI-APi-TOF show an excellent correlation with the iodic acid concentration from the nitrate CI-APi-TOF measuring in negative ion mode (see Figure 3b). Additionally, we were able to

detect iodine-containing species at higher mass to charge ratios (e.g., iodine pentoxides) during several experimental runs. Figure 7 shows the detected iodine species during an experimental run, when a high iodine concentration was injected into the chamber (mean values over a duration of 120 minutes). The derived mean iodic acid mixing ratio is ~0.98 pptv according to the measurements of the water cluster CI-APi-TOF. During this time period, we observed compounds containing up to 4 iodine atoms. The size of the circles shown in Figure 7 corresponds to the mean count rate of the signals on a logarithmic scale. For comparison, the intensities of the reagent ions are also shown. To provide more details on the observed signals, Table 2 lists the sum formulas of some identified iodine species.

During some runs, an electric field was applied to the chamber to get rid of ions and cluster ions for the study of purely neutral (i.e., uncharged) nucleation. Even during these experiments the signals as displayed in Figure 7 were present. This indicates that the water cluster CI-APi-TOF measures neutral compounds after ionizing them in the ion-molecule reaction zone. The present study only gives a short overview of the iodine signals observed so far with the water cluster CI-APi-TOF. Further CLOUD publications will focus on the chemistry of the iodine-containing species and on their role in new particle formation processes.

## 3.7 CLOUD chamber characterization

The performance of the water cluster CI-APi-TOF during measurements at CLOUD12 is shown in Figure 8. We compare the derived mixing ratios with the measurements of the PICARRO. In addition, both measurements are compared with an estimated range of ammonia mixing ratios based on the injected amount of $NH_3$ into the CLOUD chamber, the chamber volume and the ammonia life time (see, e.g., Simon et al. (2016) and Kürten et al. (2016a) for the equations linking these quantities to the estimated CLOUD mixing ratios). While the injected ammonia can easily be determined from the MFC settings, the ammonia life time in the chamber can span a wide range. For a very clean chamber or at very low temperatures the chamber walls can essentially represent a perfect sink and the ammonia has a short life time. A wall loss life time of 100 s at 12% fan speed was previously reported by Kürten et al. (2016a). Measurements of sulfuric acid at different fan speeds suggest a change of a factor of 4 in the mixing ratios when the fan speed is changed from 12% to 100%. Scaling these measurements to the ammonia measurements yields a wall loss lifetime of 25 s at 100% fan speed. On the other hand, once the walls have been exposed for a long time with ammonia they reach eventually an equilibrium where condensation and evaporation rates become balanced. Under these conditions, the ammonia life time is determined by the chamber dilution life time alone (6000 s), and so the $NH_3$ increases to higher equilibrium concentrations. Furthermore the walls can act as a source of ammonia due to re-evaporation of ammonia molecules attached to the surface. This effect can be significant when the concentrations previously injected into the chamber were higher than the current concentrations. Thus, the estimated range can vary by a factor of ~200 based on the chamber conditions. This wide range is indicated by the shaded areas in Figure 8 (light blue color).

Figure 8a shows the measurements of the water cluster CI-APi-TOF, the PICARRO, and the calculated value for ammonia. The signal measured by the water cluster CI-APi-TOF follows the injected ammonia almost instantaneously (first injection is on Oct. 23), whereas the PICARRO only shows elevated concentrations above its background of ~200 pptv much later. This

increased response time can be explained by a combination of the longer sampling line (~1.8 m compared to 1.3 m for the water cluster CI-APi-TOF), the lower flow rate (~ 1 slm with a core sampling of 5 slm compared to ~ 20 slm for the water cluster CI-APi-TOF) and the higher detection limit of the PICARRO. After the flow of ammonia is shut-off, both the mass spectrometer and the PICARRO show almost identical values when the chamber is being cleaned. Before the first ammonia injection it can also be seen that the water cluster CI-APi-TOF shows progressively lower background ammonia values.

Whether this is due to the chamber, the instrument or the sampling line becoming cleaner is unclear. Figure 8a also indicates the influence of temperature on the background ammonia level. When the chamber temperature drops from 298 K to 278 K (shortly before Oct. 31) the residual $NH_3$ decreases by around a factor of 5, which is caused by a reduction in the re-evaporating ammonia from the chamber walls. Due to the higher LOD, the PICARRO, however, can hardly detect this decrease in the VMR.

The influence of relative humidity on the gas phase concentration of ammonia is shown (time from 29.10. to 30.10.). In addition to the change in sensitivity with relative humidity shown for the water cluster CI-APi-TOF (Section 3.4), a change in humidity can lead to an increased ammonia mixing ratio in the gas phase. This is due to the fact that water molecules can displace adsorbed ammonia on surfaces (Vaittinen et al., 2014). This effect can be pronounced when the chamber walls have been conditioned with high ammonia concentrations. It is important to note that the instrument was characterized for humidity

dependency during the following CLOUD13 campaign. While changes in sensitivity with relative humidity were taken into account during CLOUD13, this was not the case during CLOUD12. The observed increase in mixing ratios at this time is a combination of a change in sensitivity of the instrument and an increase in the gas phase concentration of ammonia due to re-evaporation from the wall of the CLOUD chamber. Here, the PICARRO trace can provide insight into the magnitude of both effects indicating that the re-evaporation from the chamber walls dominates over the change in sensitivity. The time from 25.10

to 26.10 shows a steep increase in the PICARRO trace, while the ammonia trace derived from the water cluster CI-APi-TOF flattens out at 20 ppbv of ammonia. This indicates that the primary ions of the water cluster CI-APi-TOF are depleted at high vapor concentrations. It is important to mention that not only ammonia concentrations were elevated at this time, but also other vapor concentrations were rather high. During the CLOUD13 campaign, where a revised version of the ion source was used (see Section 3.2), the significant depletion of primary ions has been observed only at ammonia mixing ratios of 40 ppbv. Figure

8b shows how the ammonia mixing ratios inside the CLOUD chamber react on changes of the fan speed. The fan speed was varied between 12% (default value) and 100% to realize different wall loss rates of condensable species. As the temperature in this example is low (248 K) and the chamber is rather clean the walls act as a perfect sink for ammonia. Therefore, the measured mixing ratios almost instantaneously react to the changing fan speed indicating a change of a factor 4 in the mixing ratios. Thus, the measurements coincide with the calculated values using the wall loss life times reported above. The PICARRO

is insensitive at these low mixing ratios and cannot respond to conditions that change quickly.

### 3.8 Ammonia wall loss rates in the sampling line

The largest uncertainty in the ammonia measurement is related to the sampling line losses. At CLOUD, the sampling line is made of stainless steel and is kept as short as possible. The total length is still 1.3 m because the sampling line protrudes into the chamber over a distance of 0.5 m in order to sample air from the well-mixed center region of the chamber. Additionally, the sampling line has to bridge the thermal housing around the chamber walls. Using an ammonia diffusivity of 0.1978 cm$^2$ s$^{-1}$ (Massman, 1998) and a sample flow rate of 20 slm, the sample line penetration efficiency can be estimated as 33.7% for a laminar flow (Dunlop and Bignell, 1997; Baron and Willeke, 2001; Yokelson, 2003). This means, that if the walls of the sampling line act as a perfect sink, then the measured NH$_3$ mixing ratios would need to be corrected with a factor of ~3. However, it is quite likely that the sampling line not always acts as a perfect sink for ammonia due to desorption and re-evaporation. Furthermore, the interactions of ammonia with the surface of the sampling line depend on the humidity. Water on surfaces can affect the uptake or release of ammonia. Vaittinen et al. (2014) showed that increased humidity can displace ammonia from surfaces. Additionally, water on surfaces can allow weak acids and bases to dissociate into their conjugate compounds on the surface, thereby affecting the partitioning to the surface (Coluccia et al., 1987). Vaittinen et al. (2014) studied the adsorption of ammonia on various surfaces and found a value of $1.38 \times 10^{14}$ molecule cm$^{-2}$ for the surface coverage on stainless steel. For humid conditions this value is, however, significantly smaller and drops to ~$5 \times 10^{12}$ cm$^{-2}$ for a water vapor mixing ratio of 3500 ppmv at 278 K, which can be explained by the displacement of ammonia by water molecules (Vaittinen et al., 2014). This indicates that, depending on the ammonia mixing ratio and the gas conditions (temperature and RH), eventually an equilibrium can be reached between the gas and the surface. In such a case, no wall loss correction would be necessary. Furthermore, ammonia may re-evaporate from the inlet line walls if saturation happened at higher concentrations previously. At CLOUD, the sampling lines are attached to the chamber and cannot easily be removed during the experiments. Thus, it is not possible to quantitatively distinguish between interactions with the surface of the sampling line and the surface of the CLOUD chamber. This complicates the evaluation of the influence of the sampling line regarding the measured ammonia as a function. One practical solution would be to report average of the values considering the wall loss correction factor and neglecting the factor. When going from high values to low values, the sampling line walls can also become a source of ammonia. This can potentially lead to a strong over-estimation of the measured value and the time until a new equilibrium is reached depends on the history of the measurements.

We are aware that the sampling line losses introduce some uncertainty on the ammonia measurement. However, this is an effect other in-situ techniques also have to struggle with (see, e.g., Leifer et al. (2017)). We also want to note that the effect is much smaller for larger molecules, e.g., the penetration for triethylamine (diffusivity of 0.067 cm$^2$ s$^{-1}$, Tang et al. (2015)) reaches 61%. For atmospheric measurements, we suggest to use an inlet system where a short piece of the 1'' sampling line only takes the core sample flow from a large diameter inlet. A blower can generate a fast flow in the large inlet to essentially reduce the losses for the core flow to zero before it enters the actual sampling line (Berresheim et al., 2000).

## 3.9 Measurement of dimethylamine

As mentioned in Section 3.5 and shown in Table 1, the same calibration factor derived for ammonia was used to estimate the mixing ratio of dimethylamine. We caution, that this assumption can lead to uncertainties as the sensitivity of the measurement is expected to depend on the proton affinity of the measured substance (Hanson et al., 2011). To estimate the validity of this assumption, we compared the mixing ratios measured with the water cluster CI-APi-TOF with the calculated mixing ratios for a period when dimethylamine was actively injected into the CLOUD chamber. A chamber characterization for dimethylamine was already conducted by Simon et al. (2016), where the wall loss lifetime was determined as 432s for conditions where the chamber walls acted as a perfect sink (12% fan speed). Additionally, as discussed in Section 3.7, we use a lifetime of 108s at 100% fan speed (change in a factor of 4 when the fan speed is changed from 12% to 100%). The dilution life time during CLOUD13 is 6000 s and represents the maximum possible life time when wall loss would be negligible. Thus, the wall loss lifetime used in this study gives a lower limit for dimethylamine mixing ratios in the CLOUD chamber. Figure 9a shows the time period when dimethylamine was added. Since it takes a certain time until the stainless steel pipes of the gas dilution system are saturated with dimethylamine there is a short time delay between the switching of a valve that allows dimethylamine to enter the chamber and the rise in the measured dimethylamine mixing ratio. Once the lines are conditioned and the dimethylamine is homogenously mixed into the chamber, the measured and estimated mixing ratios are generally in good agreement with each other when the wall loss life time is used to estimate the mixing ratios. Fluctuations in the measured mixing ratio can be explained by changes in the fan speed. To estimate the consistency of the approach of scaling the calibration factor derived for ammonia to estimate dimethylamine mixing ratios, we use the ratio between the mixing ratio calculated for the Water Cluster CI-APi-TOF and the calculated mixing ratios based on the wall loss lifetime for the CLOUD chamber. For these measurements, we estimated a wall loss rate in the sampling lines of ~1.96 for dimethylamine, where a diffusivity of 0.1 $cm^2s^{-1}$ was used (Freshour et al., 2014; Simon et al., 2016). The mean deviation between the estimated dimethylamine mixing ratio and the calculated mixing ratio is 3.48 indicating that the approach of scaling the calibration factor derived for ammonia introduces uncertainties within a factor of ~3.5. The deviations at the end of the time series shown in Figure 9a are caused by nucleation experiments in which high concentrations of other vapors are used. During these stages a significant uptake of dimethylamine on particles can explain the discrepancy between measured and expected dimethylamine. Figure 9b shows a measurement of the chamber background for dimethylamine carried out during CLOUD13 over a time period of 5 days. The mean instrumental background for the time period shown in Figure 9b is ~0.14 pptv (for a temperature of 278 to 290 K and a relative humidity between ~50 and 60 %). The background values shown here are close to the background values obtained for 80% RH and 278 K (see Table 1). The observed variations are in a range of ~0.1 to 0.3 pptv provided that the measurement is not interrupted, e.g., due to the replenishment of the water source that humidifies the flow for generating the reagent ions (which explains the first drop of the background measurement in Figure 9b). The estimated detection limits shown here are below or at similar detection limits reported in previous publications (You et al., 2014; Simon et al., 2016).

## 4. Discussion and application to ambient measurements

The present study demonstrates the successful application of a water cluster CI-APi-TOF during controlled chamber experiments for ammonia, dimethylamine and iodic acid measurements. During the experiments involving iodide, neutral clusters containing up to 4 iodine atoms are detected. The technique has unprecedented low detection limits regarding the ammonia measurement as well as a fast time response and time resolution. A next step is its application to atmospheric measurements. The technique should be suitable for such measurements as the amount of clean gas required (ca. 2 slm of argon and some oxygen) is rather small and can easily be supplied with gas bottles (one argon gas bottle, 50 liters at 200 bar should last ~3 days). At CLOUD there is a restriction regarding the maximum sample flow that can be taken from the chamber. During atmospheric measurements much higher flow rates can easily be realized. Therefore, the suggested design of the inlet system using a blower and a core sample inlet should be used (see Section 3.8). Furthermore, the use of an internal calibration standard would be beneficial. We have tried to add a defined mixing ratio of $ND_3$ to the sample flow. However, besides the expected signal at ($ND_3H^+$, $m/z$ 21) further signals corresponding to $NH_4^+$, $NDH_3+$, $ND_2H_2^+$ were also visible in the spectra due to deuterium-hydrogen exchange, which makes this method unfavorable. Use of $^{15}NH_3$ for calibration is also unfavorable since the $^{15}NH_4^+$ signal is hard to distinguish from the comparatively high $H_3O^+$ signal at the same integer mass even with a high resolution mass spectrometer.

Roscioli et al. (2016) demonstrated that the addition of 1H,1H-perfluorooctylamine to the sample flow can be used to passivate an inlet, which leads to greatly reduced sampling line losses and improved time response during ammonia measurements. Recently, Pollack et al. (2019) implemented this passivation technique for ambient measurements on an aircraft. For these measurements, a tunable infrared laser was used (TILDAS-CS, Aerodyne Inc.). We also tested the passivation technique, however, the high mixing ratio of 1H,1H-perfluorooctylamine (~100 ppm to 0.1% injection into the sample flow) that is required led to a consumption of the reagent ions since 1H,1H-perfluorooctylamine has a high proton affinity and is therefore also efficiently ionized by the protonated water clusters. For this reason, the passivation technique for the measurement of ammonia can in our opinion only be used with spectroscopic techniques as it was the case in the studies by Rosciolli et al. (2016) and Pollack et al. (2019).

## 5. Summary and Conclusion

The set-up and characterization of a water cluster Chemical Ionization-Atmospheric Pressure interface-Time Of Flight mass spectrometer (CI-APi-TOF) is described. The instrument includes a new home-built cross-flow ion source operated at atmospheric pressure. The generated protonated water clusters ($(H_2O)_{n \geq 1}H_3O^+$) are used to selectively ionize compounds of high proton affinity at short reaction times. The instrument's response is linear up to a mixing ratio of at least 10 ppbv for ammonia when the derived calibration factor is applied to the normalized counts per second. The water cluster CI-APi-TOF was used at the CLOUD chamber where very low background ammonia mixing ratios (ca. 4 pptv at 278 K) were achieved.

The level of detection (LOD) was estimated as 0.5 pptv for $NH_3$. To our knowledge, such a low detection limit for ammonia measurements has not been reported so far. We attribute the low LOD mainly to the use of ultraclean argon (5.0 purity) as the

main ion source gas for the reagent ion generation. Much higher background $NH_3$ was observed when using nitrogen instead of argon. Although, the sensitivity towards the measurement of $NH_3$ depends somewhat on the relative humidity of the sample flow, the observed sensitivity changes were rather low and can be taken into account by a correction factor. We did not explicitly demonstrate the quantitative measurement of diamines (and other amines than dimethylamine) in the present study but the instrument should also be well-suited for such measurements.

During experiments involving iodine, it was observed that the protonated water clusters can also be used to detect various iodine species. A total of 29 different iodine-containing compounds were unambiguously identified, including iodic acid ($HIO_3$) and neutral clusters containing up to four iodine atoms. The water cluster CI-APi-TOF was cross-calibrated against a nitrate CI-APi-TOF measuring iodic acid during CLOUD. The two instruments showed exactly the same time-dependent trends. As there is no established calibration method for iodic acid, detection limits have been derived under the assumption

that $HIO_3$ is measured with the same efficiency as sulfuric acid, for which the nitrate CI-APi-TOF is calibrated for. The estimated LOD for the water cluster CI-APi-TOF regarding iodic acid was as low as 0.007 pptv.

Future studies will focus on the evaluation of the iodine signals and also on further signal identification in the mass spectra. Due to the instrument characteristics, we plan to apply the method to ambient atmospheric measurements to study the influence of ammonia, amines, diamines, and iodic acid on new particle formation. Airborne measurements in the upper troposphere,

where very low ammonia mixing ratios can be expected (Höpfner et al., 2016) should in principle be feasible as well. For such measurement the water cluster CI-APi-TOF technique should be very well-suited due to the very low LODs that can be realized.

**Acknowledgements**

We would like to thank CERN for the support of CLOUD with financial and technical resources. We thank the PS/SPS team from CERN for providing the CLOUD experiment with a particle beam from the proton synchroton. We also want to thank L.-P. De Menezes for providing us with a mass flow controller used during the calibrations. Next to this, we would like to thank R. Sitals, T. Keber, S. Mathot, H.E. Manninen, A. Onnela, S. K. Weber and R. Kristic for their contributions to the experiment. The discussion with Xucheng He during the creation of the paper is gratefully acknowledged. We thank C. Hüglin

for providing us with the PIARRCO and its calibration unit. This work was funded by: the German Federal Ministry of Education and Research "CLOUD-16" (no. 01LK1601A), EC Horizon 2020 MSCA-ITN "CLOUD-MOTION" (no. 764991), and EC Seventh Framework Programme MC-ITN "CLOUD-TRAIN" (no. 316662).

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

**Table 1.** Estimated limits of detection (LOD) for some compounds with high proton affinity, and for iodic acid, measured with the water cluster CI-APi-TOF. The LOD is derived by background measurements at the CLOUD chamber, where LOD = $3 \cdot \sigma$ (You et al., 2014). $\sigma$ is defined as the standard deviation of the background signal. The detection limits are based on a measurement at 278 K and 80% RH (1 minute averaging time). The measured instrumental background mixing ratios (mean values) during this time period are also indicated.

| Detected compound | LOD (pptv) | Instrumental background (pptv) | Measured $m/z$ values (Th) |
|---|---|---|---|
| $NH_3$ (ammonia) | $0.5 \pm 0.05$ | $3.73 \pm 0.35$ | 18.0338 ($NH_4^+$); 36.0444 (($H_2O)NH_4^+$) |
| $(CH_3)_2NH$ (dimethylamine)* | 0.047* | 0.058* | 46.0651 (($CH_3)_2NH_2^+$) |
| $HIO_3$ (iodic acid)** | 0.007** | < LOD** | 176.9043 (($HIO_3)H^+$); 194.9149 (($HIO_3)H_3O^+$) |

*Amine mixing ratios are estimated using the same calibration factor derived for ammonia. This can cause uncertainties. The applicability of this assumption is discussed in Section 3.9.

**Iodic acid mixing ratios are derived from an inter-comparison with a nitrate CI-APi-TOF, which evaluates $HIO_3$ based on a calibration factor derived for sulfuric acid. This assumption can lead to uncertainties but is necessary because no direct calibration method exists for such low gas phase $HIO_3$ concentrations.






**Table 2.** Iodine-containing compounds (atomic composition), together with their $m/z$ values, identified in the water cluster CI-APi-TOF spectra during the CLOUD13 campaign.

| Detected compound | $m/z$ value (Th) |
| --- | --- |
| $I^+$ | 126.9039 |
| $IO^+$ | 142.8988 |
| $HIO^+$ | 143.9067 |
| $IO_2^+$ | 158.8938 |
| $H_2IO_2^+$ | 160.9094 |
| $H_3IO_2^+$ | 161.9172 |
| $H_4IO_2^+$ | 162.9251 |
| $HIO_3^+$ | 175.8965 |
| $H_2IO_3^+$ | 176.9043 |
| $H_3IO_3^+$ | 177.9121 |
| $H_4IO_3^+$ | 178.9200 |
| $H_4IO_4^+$ | 194.9149 |
| $H_6IO_5^+$ | 212.9254 |
| $I_2^+$ | 253.8084 |
| $HI_2O_5^+$ | 334.7908 |
| $H_3I_2O_5^+$ | 336.8064 |
| $H_3I_2O_6^+$ | 352.8014 |
| $H_5I_2O_6^+$ | 354.8170 |
| $H_5I_2O_7^+$ | 370.8119 |
| $H_2I_3O_7^+$ | 494.6929 |
| $HI_3O_8^+$ | 509.6800 |
| $H_2I_3O_8^+$ | 510.6878 |
| $H_4I_3O_8^+$ | 512.7035 |
| $H_4I_3O_9^+$ | 528.6984 |
| $HI_4O_8^+$ | 636.5845 |
| $HI_4O_9^+$ | 652.5794 |
| $H_3I_4O_9^+$ | 654.5950 |
| $H_3I_4O_{10}^+$ | 670.5900 |
| $H_3I_4O_{11}^+$ | 686.5849 |

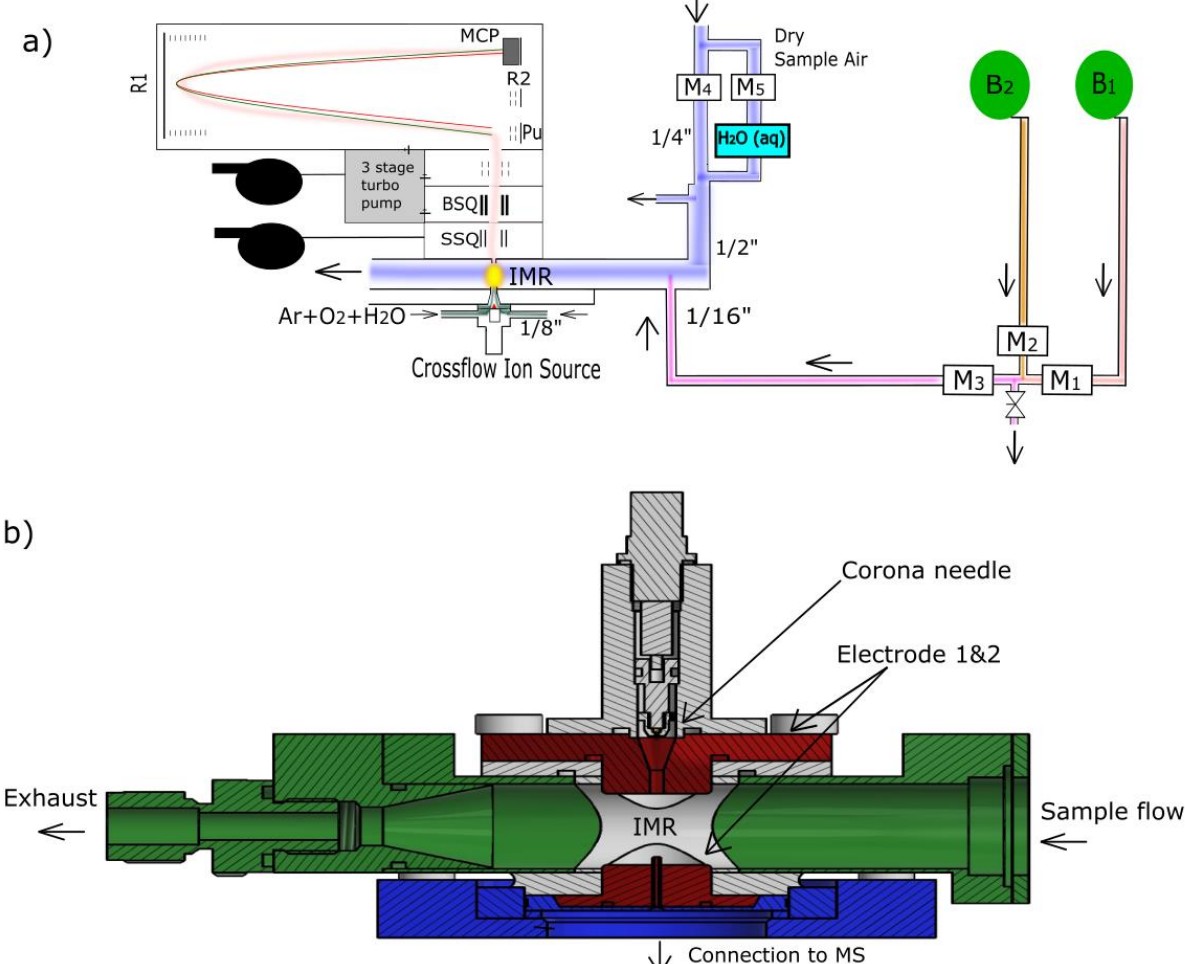


**Figure 1:** The experimental setup of the water cluster CI-APi-TOF during ammonia calibration is shown in panel a). The blue color indicates the sample flow. It consists of a mixture of 80% nitrogen and 20% oxygen. A portion of the sample flow can be humidified with a water bubbler ($H_2O$ aq) to achieve different relative humidities. $B_1$ represents the ammonia gas bottle, while $B_2$ represents a gas bottle containing pure nitrogen. There are five mass flow controllers (MFCs; labeled as $M_{1-5}$) allowing two dilution steps. Three MFCs ($M_1$, $M_2$, $M_3$) control
the amount of ammonia that is added through a 1/16'' capillary into the center of the sample flow, where the second dilution stage occurs. The reagent ions (i.e., protonated water clusters) are produced when the ion source gas (argon, oxygen, water vapor) passes a corona needle at a positive high voltage (detailed in panel b). The calibration setup is disconnected during the measurements at the CLOUD chamber to reduce backgrounds (leakage from the 1/16'' capillary). Details of the ion source used during CLOUD13 are shown in panel b. The primary ions are guided towards the sample flow using a counter electrode (Electrode 1). Additionally, a funnel is used to accelerate the primary ions
towards the sample flow. A second electrode (Electrode 2) is installed directly in front of the pinhole of the mass spectrometer. The ions enter the mass spectrometer through a capillary on the top of Electrode 2.

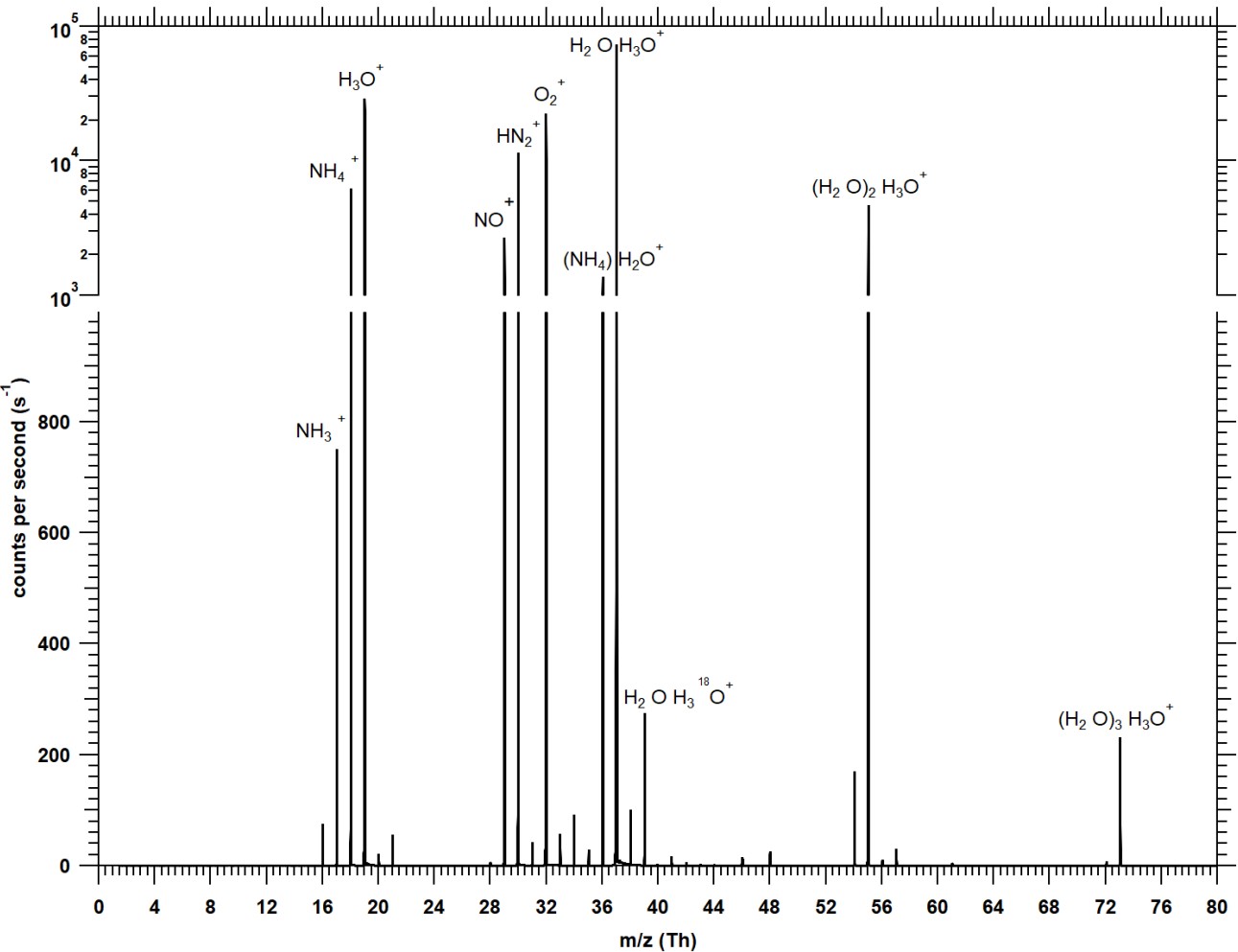


**Figure 2:** Typical mass spectrum recorded with the water cluster CI-APi-TOF when about 10 ppbv of ammonia are added during a calibration. Signals below 1000 counts per second are shown on a linear scale, while the dominant signals (> 1000 cps) are shown on a logarithmic scale. To calculate the ammonia mixing ratio, the product ion signals ($NH_4^+$ and ($H_2O)NH_4^+$) are normalized against the most

prominent reagent ion signals ($H_3O^+$, ($H_2O)H_3O^+$, ($H_2O)_2H_3O^+$, ($H_2O)_3H_3O^+$). Larger water clusters are probably also present in the ion-molecule reaction zone but a significant fraction of water evaporates upon crossing the pinhole at the atmospheric pressure interface of the instrument. Background peaks from $N_2H^+$, $NO^+$ and $O_2^+$ are always present but are neglected in the data evaluation. Due to the short reaction time (< 1 ms) in the ion-molecule reaction zone, the count rates of the reagent ions dominate the spectrum even at high ammonia mixing ratios near 1 ppbv.


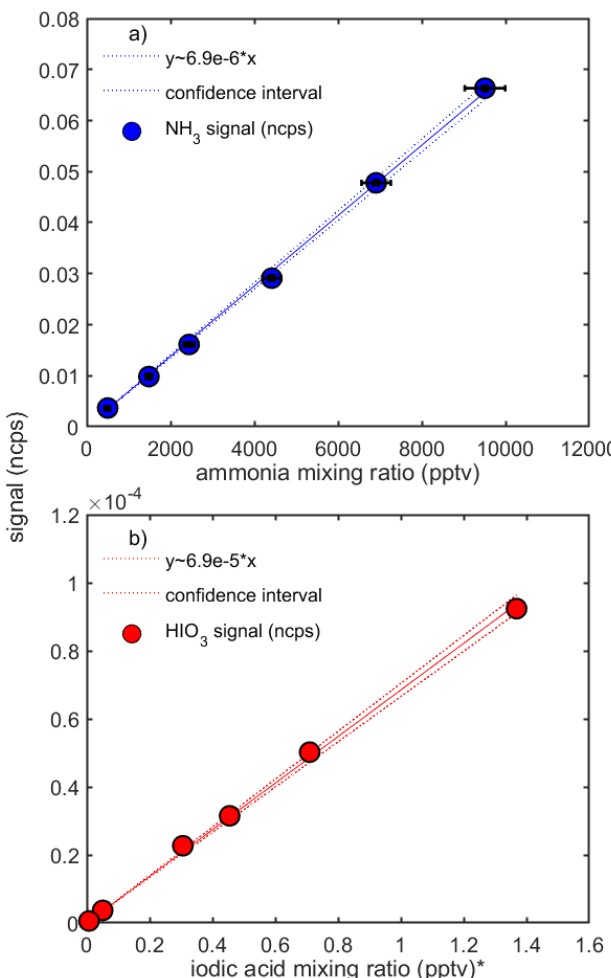

**Figure 3:** Calibration curves for ammonia (a) and iodic acid (b) at 40% relative humidity. The *y*-axes show the normalized counts per second (ncps) measured with the water cluster CI-APi-TOF. The ammonia mixing ratios are determined from the calibration set-up and the iodic acid mixing ratios are taken from simultaneous measurements with a nitrate CI-APi-TOF at the CLOUD chamber. The systematic uncertainty
of the iodic acid mixing ratios is estimated as +100%/-50% (Sipilä et al., 2016). The inverse slopes from the linear fits yield the calibration factors (see equation (1) and (2)).

*Note that the iodic acid mixing ratio is derived by applying a calibration factor for sulfuric acid to the nitrate CI-APi-TOF data.


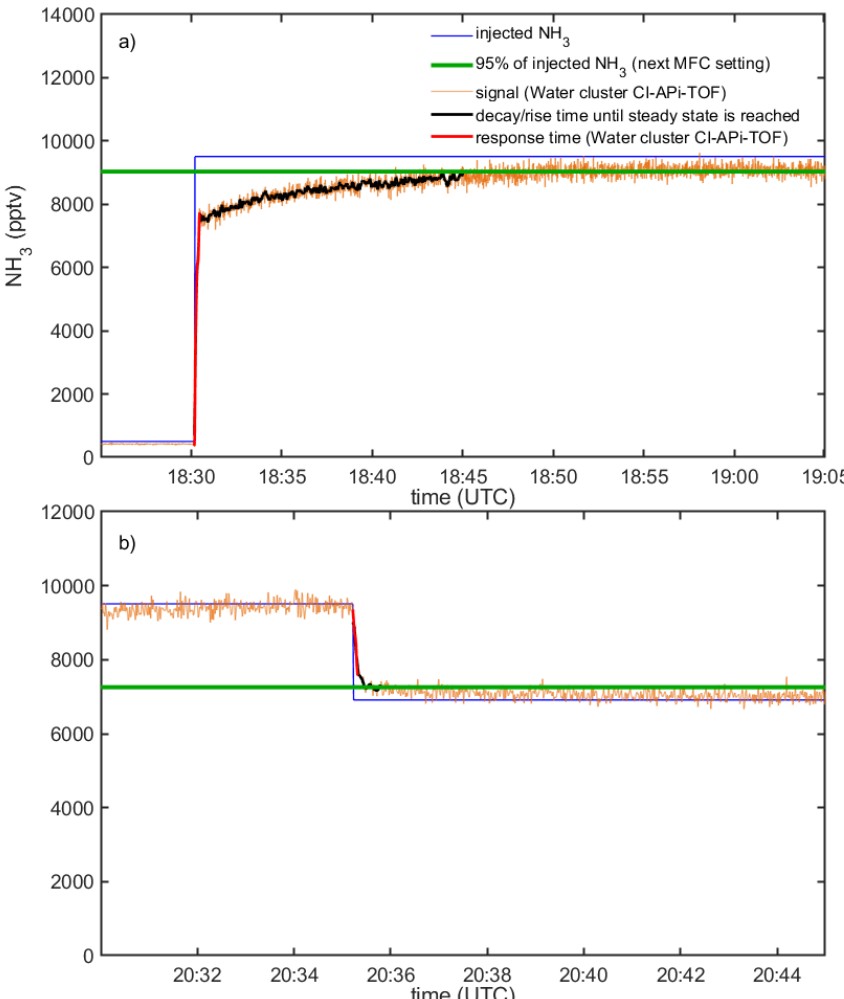


**Figure 4:** Response time of the water cluster CI-APi-TOF during calibrations at 60% RH. The injected ammonia mixing ratio from MFC settings is shown by the blue line. The signal of the water cluster CI-APi-TOF is shown by the orange line (here the data are shown with a 1 second time resolution i.e. no time-averaging is applied). The green line represents 95% of the mixing ratio being applied with the next MFC setting. The black line shows the response time until a steady state (panel a) or

95% of the final measured concentration is reached (panel b). The response time is the sum of the response time of the water-cluster CI-APi-TOF (red line) and the (slower) response time for the lines to reach a steady state where the walls are conditioned.

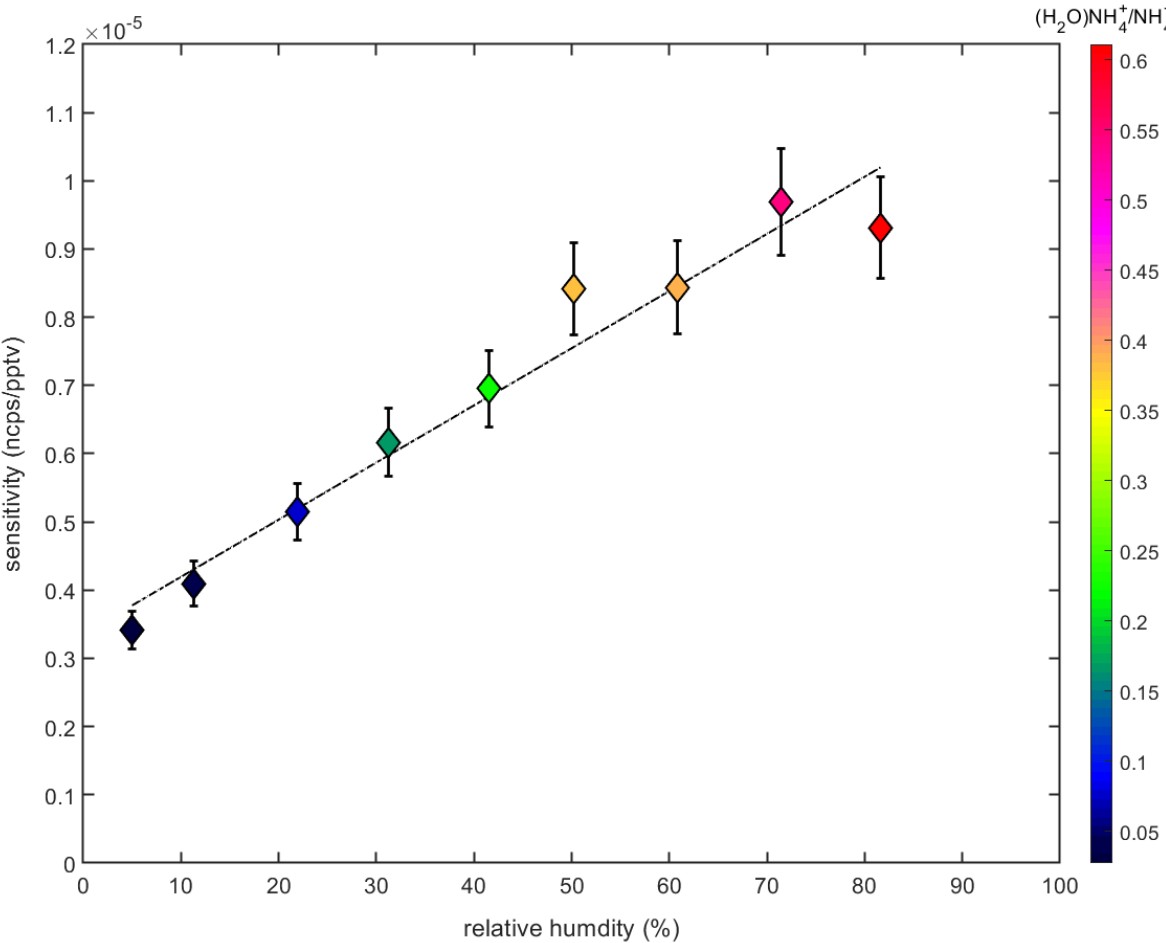


**Figure 5:** Dependency of the ammonia sensitivity as a function of the relative humidity (in %). A linear increase with relative humidity is observed, which tracks an increase of the ratio of the $(H_2O)NH_4^+$ and $NH_4^+$ ion signals (indicated by the color-code).

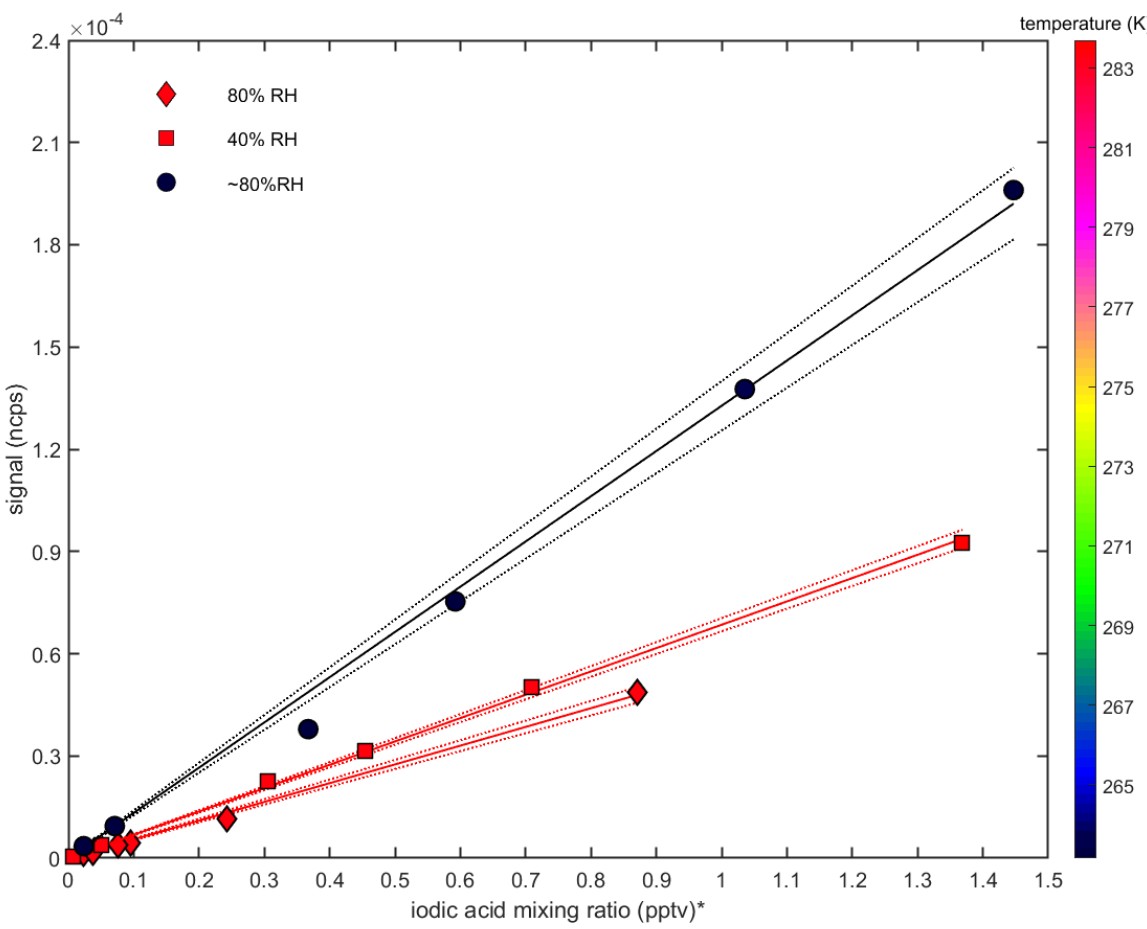


**Figure 6:** Calibration curves for iodic acid at different relative humidities and temperatures in the CLOUD chamber. The normalized counts per second (*y*-axis) are shown against the iodic acid mixing ratio measured with a nitrate CI-APi-TOF (*x*-axis). The sensitivity increases at lower temperatures, while no strong dependency on relative humidity is found at 283 K.

* Note that the iodic acid mixing ratio is derived by applying a calibration factor for sulfuric acid to the nitrate CI-APi-TOF data.

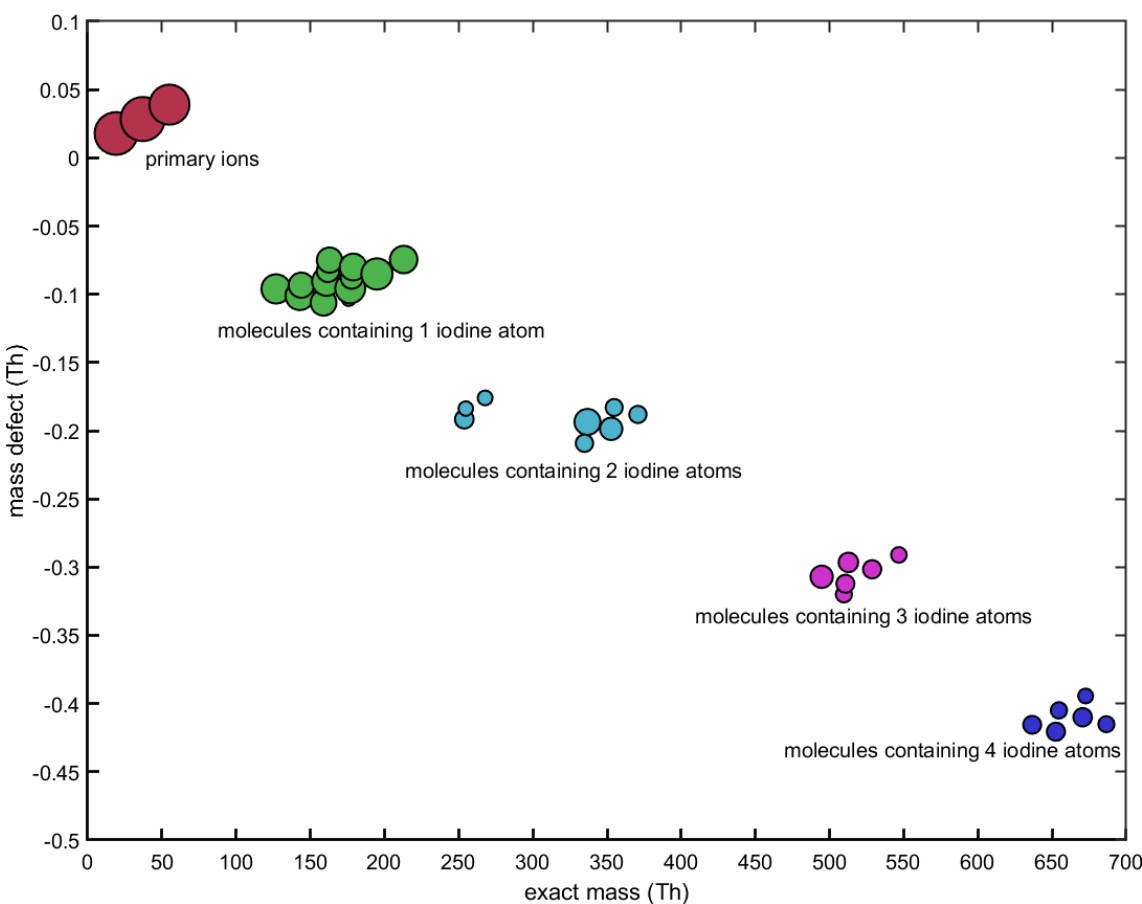

**Figure 7:** Mass defect plot for the iodine compounds, as well as the most prominent reagent ions, during a CLOUD experiment on new particle formation from iodine. The estimated iodic acid mixing ratio is ~0.98 pptv. The *y*-axis shows the mass defects of the compounds (see Table 2 and text for details), while the *x*-axis shows the absolute masses. The size of the symbols is proportional to the measured signal intensities on a logarithmic scale (from $1.24 \cdot 10^{-6}$ to 14.04 ions/s).

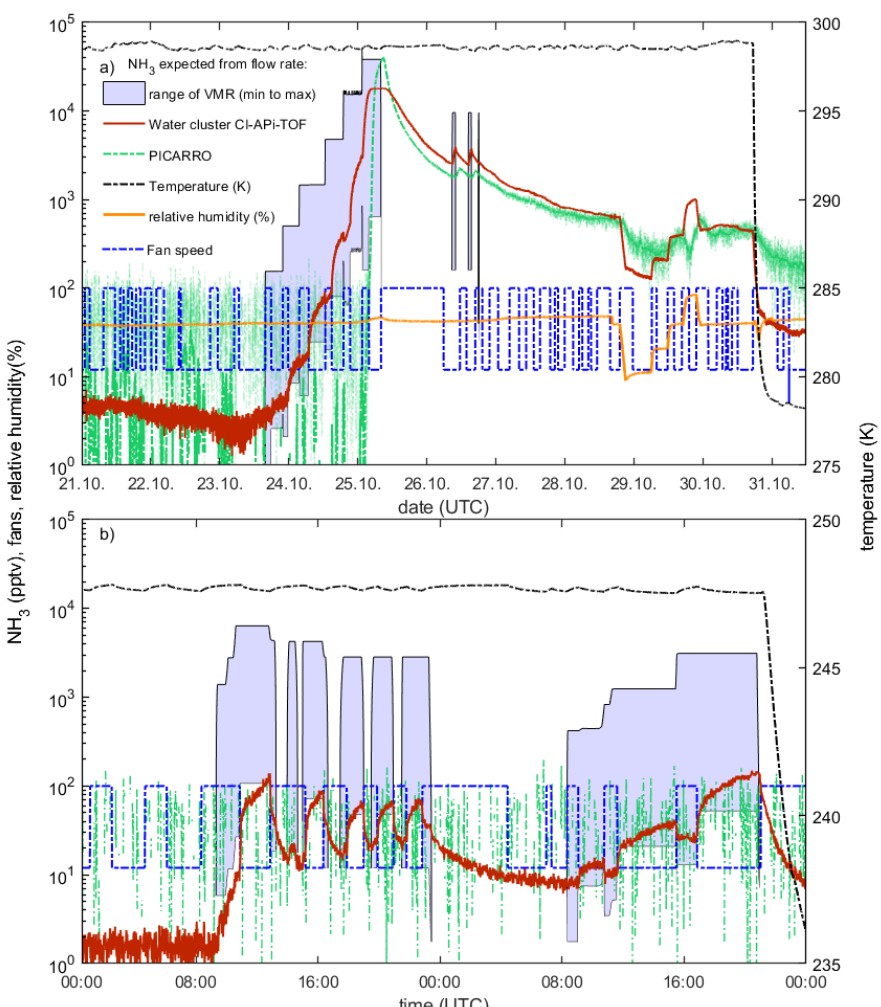

**Figure 8:** Inter-comparison between calculated (shaded blue area) and measured ammonia mixing ratios (PICARRO: solid green line; water cluster CI-APi-TOF: solid red line) at CLOUD. The PICARRO background (~200 pptv) has been subtracted, while no background was subtracted from the water cluster CI-APi-TOF. The temperature inside the chamber is indicated by the dashed black line. The speed (% of maximum, 397 revolutions per minute) of the two fans that mix the air inside the chamber is shown by the dashed blue line. The calculated ammonia mixing ratios (based on the calculated injection of ammonia into the chamber from the MFC settings) have a wide range due to uncertainties of the ammonia loss rate in the chamber. We display the maximum calculated range assuming, for the lower limit, that the chamber walls act as a perfect sink (wall loss dominated, 25s and 100 s lifetime for fan speeds 100% and 12%, respectively) and, for the upper limit, no net uptake of $NH_3$ on the walls and a loss rate determined by dilution (6000 s lifetime). For higher fan speeds, the lifetime decreases due to increased turbulence and, in turn, increased wall loss rate. Relative humidity is indicated by the orange line. The water cluster CI-APi-TOF reacts rapidly to changing conditions, such as the ammonia flow into the chamber, relative humidity, temperature or fan speed. At low concentrations, the ammonia lifetime is determined by the wall loss rate (panel b and initial stages of panel a). However, at high ammonia concentrations, the walls of the CLOUD chamber progressively become conditioned and a source of ammonia, with

corresponding increases in the ammonia lifetime and the time to reach new equilibria at lower ammonia flow rates (later stages of panel a).

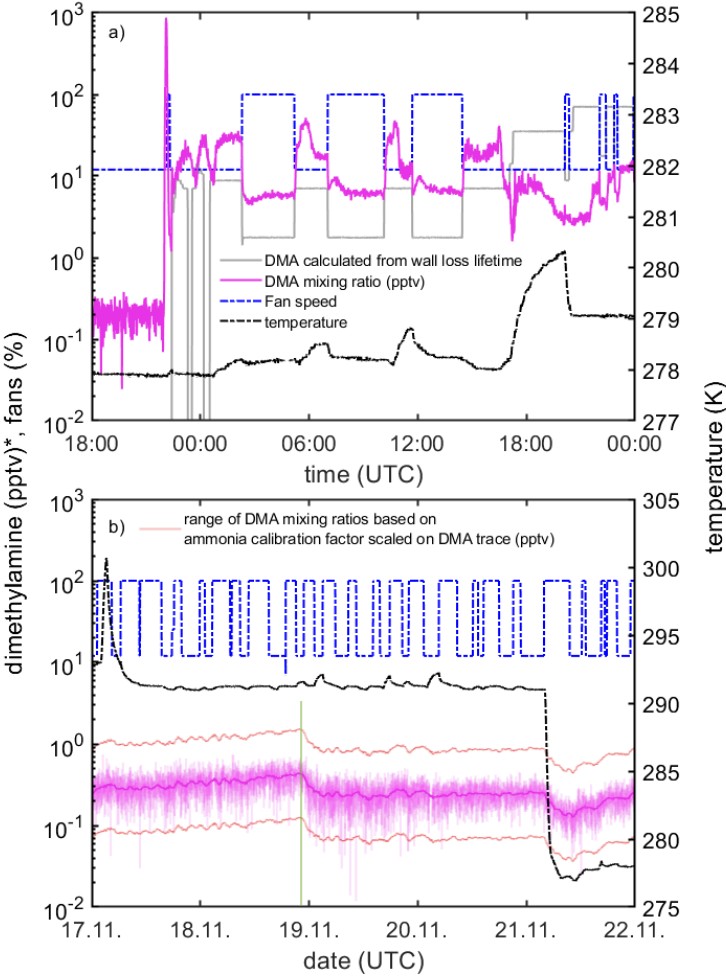

**Figure 9:** Dimethylamine mixing ratios (magenta line) during the CLOUD13 experiment. The dashed black line shows the temperature inside the CLOUD chamber. The dashed blue line shows the fan speed. Panel a) shows the dimethylamine signal during active injection into the chamber. The grey line indicates the dimethylamine mixing ratio in the chamber calculated from the MFC settings and the wall loss

lifetime. The upper limit for the uncertainty in the dimethylamine mixing ratio is a factor of ~3.5 (see text for details). Panel b) shows a measurement of background dimethylamine in the chamber over a period of 5 days, when there was zero dimethylamine flow. We consider this to be due to instrumental background and not to an actual dimethylamine background in the chamber. The thin red lines show the possible range of dimethylamine based on the scaled calibration factor (factor 3.48, 95% CL). The thick magenta line indicates a moving average of the dimethylamine background measurement. The water source has been replenished during the period shown (green line). The

mean instrumental background of dimethylamine over this period is ~0.14 pptv.

*Note that the dimethylamine mixing ratio is determined with the calibration factor for ammonia.