# Peer review of "Measurement of ammonia, amines and iodine species using protonated water cluster chemical ionization mass spectrometry"

_Atmospheric Measurement Techniques, 2019_

## Referee Comment (RC1) · Anonymous Referee #1 · 15 Aug 2019

General Comments

This paper presents the use of H3O+ reagent ion chemistry on a CI-APi-TOF mass spectrometer to detect ammonia, dimethylamine, and iodine oxides in the CLOUD chamber at CERN. The detection of ammonia is pushed further into a quantification technique by performing calibration with a primary standard and intercomparing with a Picarro NH3 analyzer. The calibration response is characterized for relative humidity dependence since the water cluster ions increase in importance and result in a higher sensitivity. The calibration is then demonstrated to result in a reasonable capability to quantitatively measure NH3 with the CI-APi-TOF in the CLOUD chamber, within the

expected range of mixing ratios from a set of standard addition experiments. The detection of the other species is presented and their detection limits estimated based on parameterizations with other instrument responses or assumptions on the H3O+ ionisation efficiency. Overall, this work presents an interesting new method for quantifying experimental NH3 concentrations that may be possible to translate to atmospheric observations, which is a suitable contribution to Atmospheric Measurement Techniques. However, there are several major issues in the estimated detection limits for dimethylamine and iodic acid and Picarro intercomparison that must be addressed with rigour or removed from the manuscript in order for it to be acceptable for publication.

Major Comments

1. There could be better clarification throughout the manuscript to indicate that the amine and iodine species are detected and that their mixing ratios are estimated semi-quantitatively, as formal calibrations have not been performed with a primary standard. The title should be revised to reflect quantitation of ammonia and detection of DMA and iodine oxide species specifically instead of 'amines and iodine species' which is misleading. The discussion clearly states that the DMA and HIO3 quantities are estimates based off of scaled numbers from H3O+ or NO3- CI-APi-TOF responses to other species (e.g. NH3 or H2SO4). Converting the units from ncps to mixing ratios for those scaled responses leads to values that may be highly inaccurate. The potential for quantification and sensitive detection of these species by H3O+ CI-APi-TOF can be discussed, but the scaled response estimates should not be used to report mixing ratios, such as in Figures 5 and 8. These should be replaced with signal reported as normalized counts per second (ncps). The authors state in several locations in the discussion potential sources of bias from the assumptions made in converting signal to mixing ratios, which are not fully characterized for the CLOUD chamber and could easily represent a factor of 2 or more error. Based on the presented data, mixing ratios for these compounds should be replaced with ncps units throughout the manuscript. The authors make arguments for the validity of some assumptions throughout the manuscript

as well, which seem to be undermined by other parts of the discussion. See the technical comments below for instances where this is the case and the discussion should be revisited.

2. Intercomparison with the Picarro NH3 cavity ring down instrument does not evaluate the measurements correctly and ignores discussion regarding the limitations of the CI-APi-TOF approach (e.g. response times on the order of hours) or the experimental setup of the intercomparison (e.g. the Picarro connected to the exhaust of the CI-APi-TOF). For example, there is a clear background offset in the Picarro measurements from the chamber that is not accounted for, but for which the CI-APi-TOF dataset is corrected accordingly through its independent calibration. There are other important measurement concepts that are missed in the data analysis here such as the aforementioned background offset correction, inlet and instrument surface sorption/desorption effects impact on response times, and detection limits. See the technical comments corresponding to these sections of the manuscript below for several specific comments. It is possible that the intercomparison is technically invalid based on the experimental setup and should be removed from the manuscript, but this cannot be assessed without further data provided by the authors regarding the setup of flows to direct the chamber air to the instruments.

Technical Comments

Page 1, Line 17: Diamines are not presented in the manuscript and are mentioned here. Remove. There seem to be several issues of material that was planned for inclusion in this manuscript that have been removed, but not in all locations. The authors should revisit the manuscript with this in mind to improve clarity throughout.

Page 1, Lines 20-21: The detection limit difference for the amines should be stated as 'at least XX times lower' here to be more specific.

Page 2, Line 33: 2.5 is usually a subscript

Page 2, Line 36: Use one or the other of 'new particle formation' or 'nucleation' and add 'e.g.' before each example of the ternary and multi-component systems.

Page 2, Lines 40-46: This clarity of this section is not very good because these sentences are discussing too many topics, which should be separate sentences. Delete 'their' and end the sentence after the Dunne et al. reference. In line 42, replace 'show' with 'support this by observing'. On line 44 end the sentence after 'water' and start the next sentence with 'For example' instead of using 'e.g.' in-line.

Page 2, Line 46: It is accurate to state 'enhance nucleation rates'. The 'even stronger' is not necessary and confuses the sentence's evaluation of relative enhancements of amines over ammonia in forming new particles.

Page 2, Line 48: Why have the authors stated 'in principle confirmed' here? Perhaps the specific property assessed (e.g. thermodynamically favored formation) would be a more accurate term to place here.

Page 2, Lines 52-53: The list of references for amine mixing ratios is more comprehensively summarized in a review by Ge et al. in Atmos. Environ. (2010) which can replace this list of references[1].

Page 2, Lines 54-57: There is an attempted motivation here that CI-APi-TOF is more versatile than other analytical techniques for atmospheric samples of reduced nitrogen species. However, separation techniques are more selective than CI-APi-TOF, particularly when it comes to the detection of structural isomers. Optical absorption techniques by systems such as cavity ringdown and quantum cascade laser systems are easily as sensitive and with as high time resolution as CI-APi-TOF. The limitation of the prior methods that CI-APi-TOF overcomes should be made with greater clarity here in order to motivate the CI-APi-TOF analytical approach. It would be easiest to remove the first sentence of this paragraph and start the second sentence at 'Chemical ionisation atmospheric pressure interface mass spectrometry. . .'

Page 6, Lines 167-171: This is a big stretch in justification for quantifying amines. Strong acids detected by CIMS do not have the same response factors per mixing ratio detected and this is well reported in the literature even though their reaction rates would be predicted to be the same for a given ionisation technique (e.g. 2-16 counts per pptv for strong acids by proton exchange in Roberts et al. in Atmos. Meas. Tech. (2010)) [2]. This detail should be clarified here along with the potential outcome of the assumption for amines likely being up to a factor of 10 error in quantitation as a worst-case outcome. It is likely better to report in this work the measurement of amines only in 'ncps' and to do a ballpark estimation of the mixing ratios in the discussion.

Page 6, Line 186: A sentence should not begin with an acronym. Rephrase.

Page 7, Lines 207-209: This is quite the string of assumptions and underscores the major issue with claimed quantitation in this work. The detection of HIO3 is, at absolute best, qualitative and units should not be assigned to the measurement in pptv, but 'ncps'.

Page 7, Lines 209-212: This intercomparison is shown in Figure 5, but the axis labels do not communicate this. They should both be reported as normalized counts since the NO3- CI-APi-TOF signal is scaled on the H2SO4 response, which does not yield a quantitative calibration factor for HIO3, but only a relative signal.

Page 8, Lines 223-226: The authors state that the instrument was independently calibrated. This means that they have data to calculate the detection limit of the Picarro NH3 cavity ringdown system and it should be done here instead of stating the value given from the datasheet.

Page 8, Line 237: The first name of the author should not be given here.

Page 8, Lines 248-251: This comparison of the stability of reagent ion should be made by calculating the change in the calibration slope (ncps vs NH3 mixing ratio) between high (1-10 ppbv) and low (<1 ppbv) ranges to reach the conclusion that the sensitivity

is consistent across the full range of experimental mixing ratios used in CLOUD. The authors also state 'sufficient precision' here without defining what the value they use is. The numeric value should be given. It would also be useful here for the authors to comment on the likelihood that this calibration response will hold up under ambient observations where other atmospheric components will compete for H+ transfer.

Page 9, Lines 259-261: Here is the HIO3 intercomparison again. The error in the VMR is estimated based on an applied scalar, which is effectively a guess. It is more transparent, and valuable, to report that the two independent ionisation schemes yields a strong linear response, supporting the sensitive detection of HIO3 using H3O+ chemistry. Extending this further into estimated mixing ratios is not justified.

Also, for these calibrations, the experiments shown in the remainder of the manuscript are not collecting 20 minute time-resolution data. Averaging at timescales more relevant to the measurement timescale would give a more accurate estimate of calibration response and quantitation accuracy, so long as 20-30 data points are pooled for each calibration mixing ratio.

Page 9, Lines 263-264: The resulting difference in the forced and unforced slopes should be reported with the numeric value of the percent difference.

Page 9, Lines 265-267: 'concentration steps' should be 'calibration mixing ratios' and 'confidence bounds (95 % confidence intervals)' should be '95 % confidence intervals'.

Page 9, Lines 272-274: The response time of the CI-APi-TOF to a stable signal following the stepped-down changes in NH3 VMR should be provided here explicitly. The discussion following this section states that the instrument and line surfaces contribute to background signals observed. Here, the 'diffusion of ammonia from the capillary into the sampling line' should be clarified to indicate that the background observed also has contributions from all surfaces in the calibration system, the instrument inlet, and the instrument walls.

Pages 9-10, Lines 282-286: The changes in calibration factor should be explicitly given. Are they a factor of 2 different or a factor of 5? This is a measurement technique manuscript and it is VERY important to highlight how small changes in instrument operation and setup can affect the quality of the measurements. Was the CI-APi-TOF re-calibrated with these slight changes? How was the calibration factor derived for these CLOUD experiments? Did the response time of the inlet change? All of this information has high value here.

Page 10, Line 294: What does 'including all components' mean?

Page 10, Lines 301-306: This section of discussion is unclear and difficult to follow. Rewrite for clarity since looking at Figure 7, there seems to be disagreement with what is stated here regarding 15 % and 3 % change in signal. Perhaps there is a way to depict this more clearly in a new figure?

Page 10, Lines 308-310: Rewrite this sentence, removing all information in brackets and either placing it explicitly in the sentence or removing it. Also, should the reference to Figure 4 here actually be to Figure 5?

Page 11, Section 3.4: Detection limit calculations require a stable background signal when the instrument is known to be sampling a negative control. Quite a bit of this section discusses contamination issues, which should be made into its own section and kept separate from detection limits.

Page 11, Lines 326-327: These facts should be moved to the calibration section as an addition to the statement about the capillary NH3 contamination since all of these components can be causing the described time lag in the calibrations and the observations.

Page 11, Lines 334-335: This is a sentence fragment and does not belong in the part of the discussion. It should be moved to the discussion of the RH-dependent sensitivity along with context driven by the data presented in Figure 4.

Page 11, Lines 336-340: Giving approximated detection limits, along with the assumptions being made for HIO3 and the amines in the discussion is alright, so long as it is very clear that these are initial guesses. Where these numbers absolutely do not belong is in Table 1, which should be removed from the manuscript. It is highly likely that these numbers will be used out of context and with disregard for the assumptions made here for the estimation (e.g. not everyone will have a NO3- CI-APi-TOF calibrated for H2SO4 to scale their measurement against for HIO3, along with the assumption that the sensitivity is equal on top of that). The authors need to be clear throughout this work that the estimated mixing ratios are consistent with expectations, BUT that a calibration with known quantities of the target analytes should be performed by anyone wanting to make quantitative measurements of these compounds. Again, any figures showing mixing ratios or amines or HIO3 should be converted back to units of ncps since these estimates are based on tenuous assumptions. Keeping some mention of the potential detection limits in the discussion is alright as it motivates further work in using this instrumental platform for quantitative analysis.

For this reason, the title of the manuscript needs to be revised to reflect what species can be quantified and which can only be detected. Without calibration from a primary standard, HIO3 and amines are only detected in this work, while NH3 is quantified and this should be kept clear.

Page 11, Lines 341-347: This part of the discussion is quite unclear. 'E.g.' should be 'For example,'. What is the peak with the highest count rate for the amines and why does this avoid interference for the mentioned fragments? This is not adequately explained. In the final two sentences, the authors undermine their assumptions regarding the detection of amines identically as they detect ammonia, stating that they omit the use of larger product ions, which will skew the sensitivity of detection. Then, this bias is dismissed as negligible because the water cluster signals are smaller for the bases than ammonia. Clearly, this indicates different reagent ion chemistry and product distribution for the amines relative to ammonia, which is inconsistent with assumption that

they are the same and that the NH3 calibration factor can be applied to the calculation of amine mixing ratios and detection limits. This is strong evidence against reporting mixing ratios and detection limits for C2-amines in this work. The strength of this paper lies in the quantitative detection of NH3 and show focus on that data, with discussion of these additional species kept to a minimum.

Page 12, Line 361: 'Thus, the sensitivity...' this sentence is redundant with the preceding discussion and can be removed. This paragraph and the one preceding are nice guides to some of the technical considerations for measuring these target analytes and the controls on signal/noise for each of the ions. The discussion of LOD in these two paragraphs for amines and iodic acid should be changed to discuss signal-to-noise of the instrument.

Page 12, Line 372: 'up to the tetramer (...' could easily be replaced with 'molecules containing 1-4 iodine atoms' and be much more accurate. This should be changed throughout the manuscript, including Figure 6. Calling these species monomers, dimers, etc suggests that the rest of the atoms are part of a fundamental subunit one would typically find in describing a polymer, which does not seem to be the case here? If there are fundamental subunits for these I-containing species and the terminology of monomers, dimers, etc holds, then this should be more clearly written to specify the atoms found in remainder of the subunit.

Page 13, Line 395: The accuracy of the chamber concentrations being much less certain than those from the measurements of NH3 by the CI-APi-TOF should be used earlier on to motivate the need for this instrument being characterized fully.

Page 13, Lines 400-401: Could this difference in response time during NH3 mixing ratio increases be due to inlet location differences between the instruments inside the chamber? Are the different responses due to concentration gradients as NH3 mixes throughout the chamber? Or are there very different inlet line lengths? The intercomparison description on Page 7, Lines 219-223 are insufficient to determine if these instrumental measurements are even capable of being properly compared. The authors suggest that in 'some intercomparison measurements' that the Picarro was hooked up to the exhaust of the CI-APi-TOF instead of sharing the same sampling line. Surely there are huge losses of NH3 inside the CI-APi-TOF such that it is unreasonable to expect quantitative transfer of NH3 out of the instrument exhaust? This type of setup would dramatically bias an intercomparison and may invalidate its applicability altogether in this work. The authors should re-evaluate whether the experimental setup was sufficiently robust to expect NH3 to be transmitted to both instruments in a manner that does not compromise the sample composition. If the conclusion is that this is not possible, then this section of the manuscript needs to be removed as it is likely invalid.

Also, why was a background correction not applied to the Picarro data based on having it sample zero air? The Picarro gas analyzers typically arrive from the factory with a calibration factor and offset applied and can be much more sensitive than the operators' specifications sheets. Given that an independent calibration was done using a permeation device and, presumably an overflow with zero air, the instrument detection limits should have been possible to independently calculate, along with any systematic offset in the instrument response, which could be corrected where this is identified as a factory offset. The calibration of this instrument with external NH3 would have allowed the LOD for the instrument to be determined the same was as it was for the CI-APi-TOF (i.e. using S/N=3 from the blank observations and comparing that to the background measurement). Making the intercomparison with more consistent treatment of the calibrations performed on both instruments would strengthen the discussion throughout this portion of the manuscript.

Page 14, Line 417: The material used for the sampling line is not discussed. Part of the discussion from literature line losses (Line 426) talks about stainless steel, but this would be atypical for an NH3 sampling inlet. The authors need to clarify this material so the context can be properly evaluated. Given the incredibly long decay time constants from the chamber (on the order of days according to Figure 7?), which

are also not presented for the instrument from the calibrations, it is hard to determine the relative order of the effects here. See comments on the calibration figures for some improvements to the manuscript with additional figures that could be made.

Page 14, Lines 426-434: The water effects on line losses seem to be ignoring an established effect from the literature. Water on inlet surfaces can allow weak acids and bases to dissociate into their conjugate compounds on the surface, increasing the partitioning to the surface. Adsorption on surface sites alone is too simple of an interpretation. Further to this, data is not presented that decouples the effect of the inlet lines from the desorption of NH3 from the CLOUD chamber walls. The value for inlet desorption is only described as 'it can take a long time'. It would be nice to see some form of time constant determined for the inlet to return to initial conditions and a reflection on whether this waiting period is reasonable compared to replacing the lines between experiments.

Page 14, Line 441: All of Section 3.8 needs to be rewritten in terms of signal instead of mixing ratios, except where estimates are discussed. Figures referred to in this section should all report signal using units of ncps since a direct calibration has not been performed. The comments on how the estimated calibration factor and the observations in the chamber are consistent within the factor of 10-100 uncertainty for DMA put in the chamber can be retained to demonstrate that the assumptions for the CI-APi-TOF calibration factor are also likely within this same factor of 10-100. In particular, the line of reasoning on Page 15 from Lines 461-464 is based on biased expectations where an arbitrary correction estimated at a value of 2 would bring the observations 'into even better agreement' with the expected mixing ratios, but this does not have much factual basis since the concentrations shown in Figure 8 span an order of magnitude depending on wall losses and chamber dilution. Perhaps stating that the observations fall within the expected range of values is sufficient for this section and the speculation on correction factors can be removed since a direct calibration of DMA was not actually performed.

Page 15, Line 473: The specific analytes discussed are ammonia, dimethylamine, and iodic acid and this should be specified here. Depending on the actual chemical species comprising the remaining iodine oxides (or up to tetramers of iodic acid if this is the case), the iodine species detected can be made more chemically specific in the conclusion. Based on Table 2, iodine oxides seems to me to be the best term, and it could be used effectively throughout the manuscript.

Page 15, Line 474: Neither fast time response or time resolution values are given in the manuscript. Indeed, they could yield wonderful proof of the developed technique and should be included. See comments on Figures with some suggestions.

Page 16, Lines 488-491: In addition to having high proton affinity, this perfluorinated amine is also a strong surfactant which promotes its ionisation further.

Page 16, Conclusions: Revise in light of all other changes to the manuscript.

Page 27, Table 1: Remove. The only true detection limit measured is for NH3 and the rest are estimated based on assumptions that may have significant error. Further to this, the LODs were acquired over 2.5 hours of signal acquisition, while the measurements are reported at 1 minute intervals. A more accurate assessment of detection limits would have been determined at similar timescales, using about 20-30 background measurements.

Page 29, Figure 1: Numeric or alphabetic labeling of the instrument parts with corresponding descriptions in the caption would make this diagram easier to follow as a lot of the text is obscured by color or very small. The 'argon+oxygen+water vapour' could be replaced with 'Ar + O2 + H2O'.

Page 31, Figure 3: If a fit is forced through the origin, then a measure of instrument signal while ultra pure zero air should be included as a blank to determine if there is an offset in detection, particularly for NH3. For the HIO3 measurement, the x-axis is a scaled value based off of H2SO4 detection and not a true measure of HIO3 and a

comparison of ncps would be more useful to describe the sensitivity of the H3O+ CI-APi-TOF versus the NO3-. For this second plot, was any background value subtracted?

An important component of instrument performance that can be evaluated while doing calibrations such as these is instrument response times (both with increasing and decreasing analyte concentration) from 0-95 % of max signal and 100-5 %. The authors claim that the instrument has a rapid time response with high time resolution, yet no such data is presented in figures and no numeric values reported. This is critical to report as it also helps provide clear context in the interpretation of the chamber observations.

Page 32, Figure 4: Was this sensitivity dependence on RH used to correct the dataset shown from the chamber? With the changes in temperature, if there is water in the chamber, then there will be a change in sensitivity that should be applied and may change the interpretation of the datasets.

Page 33, Figure 5: The x-axis should also be signal-based units. The caption states that there is no RH dependence, yet there is a clear difference within the regressed data that shows this and suggests that it is also statistically significant between 40 to 80 % RH (color suggests that T is constant for these), and consistent with the findings for detection of NH3 increasing in sensitivity with increasing RH.

Page 34, Figure 6: The logarithmic scale for the marker size should be provided here. The nomenclature replacing 'monomers', etc. should be revised as necessary.

Page 35, Figure 7: The two data components on the y-axis should be separated with a ',' and not with a '/' which implies division. The font size of everything in this figure needs to be made larger. Depending on the outcome of the details of the intercomparison assessment, the Picarro data may need to be removed from this figure, especially if it was connected to the exhaust line of the CI-APi-TOF here. Panels a) and b) are not described in the caption and should be. An additional panel that may be of use in the intercomparison is a regression of the CI-APi-TOF measurement against that of

the Picarro when both detection limits are properly accounted for (i.e. properly background corrected based on calibrations that were performed). In the case where one instrument has a higher detection limit than the other (which seem likely to be the case given the sensitivity of the CI-APi-TOF), then the higher LOD should be used as the cut-off for the intercomparison.

Page 36, Figure 8: Add a marker where the H3O+ source water was changed on panel a) and make a note of it in the caption. Convert the mixing ratio to ncps and comment on the expected range of values with an assumed calibration constant in the caption. It is surprising that a direct calibration of DMA was not performed since permeation devices for this compound are commercially available, as they are for NH3.

References

[1] X. Ge, A.S. Wexler, S.L. Clegg, Atmospheric amines e Part I . A review, Atmos. Environ. 45 (2011) 524–546. doi:10.1016/j.atmosenv.2010.10.012.

[2] J.M. Roberts, P. Veres, C. Warneke, J. a. Neuman, R. a. Washenfelder, S.S. Brown, M. Baasandorj, J.B. Burkholder, I.R. Burling, T.J. Johnson, R.J. Yokelson, J. De Gouw, Measurement of HONO, HNCO, and other inorganic acids by negative-ion proton-transfer chemical-ionization mass spectrometry (NI-PT-CIMS): Application to biomass burning emissions, Atmos. Meas. Tech. 3 (2010) 981–990. doi:10.5194/amt-3-981-2010.

---

## Referee Comment (RC2) · Anonymous Referee #2 · 17 Sep 2019

The manuscript "Measurement of ammonia, amines and iodine species using protonated water cluster chemical ionization mass spectrometry" describes and discusses a new technique using protonated water clusters (H3O+) with a Chemical Ionization-Atmospheric Pressure interface-Time Of Flight mass spectrometer (CI-APi-TOF) to detect ammonia, dimethylamine, and iodine oxides during experiments performed at the CERN CLOUD chamber. The technique is calibrated for ammonia using a primary bottle standard and dilution system. It is also compared to a commercial cavity-ring down spectrometer measuring ammonia and water vapor manufactured by PICARRO, Inc. No direct calibration is presented for the other species detected, rather the calibration factor for dimethylamine is assumed to be the same as for ammonia and that for

iodic acid is parameterized from the observations of a nitrate CI-APi-TOF instrument making simultaneous measurements in the chamber. Humidity effects on the ammonia sensitivity are investigated. Limit of detection is determined. A characterization of instrument as operated at the CLOUD chamber is presented. The subject matter of the manuscript is appropriate for Atmospheric Measurement Techniques.

Though I do find it exciting that this type of ion source for CIMS instruments is being developed and it shows much promise, I do feel that the manuscript glosses over significant details for evaluation and does not provide enough justification of others to be published without revision.

Major comments: 1. Though powerful, CIMS is not an absolute measurement technique. A good, defensible calibration is necessary. The manuscript should do a better and clearer job indicating that only the ammonia detection is calibrated with a primary standard and that the mixing ratios for the other species are estimated qualitatively. It is too easy for the reader to lose sight of this fact, since no differentiation between calibrated and estimated mixing ratio results is made in the table or the figures.

2. The main product of this work is the development of the ion source. However, details of the ion source are lacking in the text and figures. Figure 1 is more of a cartoon than a schematic. The details that are given in the text (page 5, lines 143 – page 6, 162) are difficult to translate to Figure 1. For example, a counter electrode and capillary are described in the text but not identified in the figure. Dimensions are given in the text that are not shown in the figure. This makes it unnecessarily difficult for the reader to follow how the ion source truly works and evaluate its perfromance. Also, details such as tubing length and flow rates for the calibration dilution components should also be given.

3. The comparison of the LOD and low background for this water cluster CI-APi-TOF instrument to others is not as straight forward as presented here. Here the calculated detection limits are based on a 2.5 hour measurement of synthetic air generated from

liquid nitrogen and oxygen with a 1 minute average for a single data point (See 2.1 and Table 1). This leads to the question are the values given in Table 1 those after sampling the synthetic air for 2.5 hrs or the values for the full 2.5 hr time period? If is the latter, what was the time required, if any, for the signal to drop to the 3 pptv level after removal of a 5-10 ppbv ammonia calibration addition?

Unfortunately, no data is presented to support the low background and LOD claim. The time series in Figure 7 is not applicable because the effects of the CLOUD chamber cannot be separated from those of the instrument. Here a time series showing the addition of ammonia and the instrument response to its removal in the set-up shown in Figure 1 would be extremely useful. This would also better mimic field measurements, for example, measurements at a ground site when wind shifts from a region with ammonia sources to one without. Then a better comparison could be made to other instruments and their field measurements. Other factors, affecting signal stability, i.e., LOD, include vibrations, for instruments on mobile platforms such as vehicles or aircraft, and heat, for instruments in trailers, on towers, in vehicles, and in aircraft. In many ways the controlled laboratory conditions associated with the CLOUD chamber provide as ideal environment. While the work presented here is impressive, care should be taking comparing the performance there to that reported in a field campaign. This also highlights the necessity of evaluating instrument performance in-situ for every campaign and not relying on spec sheets or one laboratory test.

4. No data is shown to support that this is a fast time resolution measurement either. Similar to my previous comment, a time series showing the signal decay after removal of ammonia would be helpful in evaluating the time response of the instrument. Also, if this is a fast time resolution measurement why does it take at least 20 minutes (Page9, line 254) for the signal to reach the mean value of a steady state measurement used in the calibration curve shown in Figure 3a?

Specific Comments Page 1 line 17, If the authors did not explicitly demonstrate the quantitative measurement of diamines (see page 16 line 504) then they should not be

mentioned in the abstract. Similarly, amines should be changed to dimethylamine since that is the only amine for which data is shown. Speculative future application should be saved for the discussion in the manuscript not the abstract.

Page 1 line 22, Classifying 10 ppbv ammonia as high is very subjective. What is high for the CLOUD experiment may be typical or even low for many areas as seen in many of the publications cited in this manuscript.

Page 5 line 151, Please show how the reaction time is estimated here.

Page 5, line 158, Are all tubing diameters given in the manuscript representing the outer diameter? The inner diameter should also be given as that affects the flow characteristics.

Page 6, line 180, What is the uncertainty in the mixing ratio of the ammonia bottle and who is the manufacturer? It should be given here with the calibration set-up details not later in the manuscript discussing the calibration results.

Page 7 line 199, What is meant by 'a fairly short equilibration time'? Minutes, hours, seconds? Please show the time series of the ammonia signal as a function of the step changes in ammonia added in addition to the calibration curve shown in Figure 3.

Page 7, line 209-210, How is the assumption given here that both sulfuric and iodic acid are detected with the same efficiency by the nitrate CI-APi-TOF justified? Wouldn't this make the estimated iodic mixing ratio a limit in some regard?

Page 7, line 219, The text mentions the PICARRO being connected to the exhaust line of the water cluster CI-APi-TOF for comparison. Is this comparison discussed or shown in the manuscript? If not, why? Or am I incorrect that Figure 7 is showing PICARRO measurements being made from its own sampling line on the CLOUD chamber and the comparison is shown in Figure 7? The text also mentions tests when the flow is increased to the PICARRO. When is this used? If used when the PICARRO is sampling the exhaust of the water cluster CI-APi-TOF is the flow to the water cluster

CI-APi-TOF increased also? Again, another instance lacking enough detail to evaluate the experiment and experimental results.

Page 9, line 264, I am confused by the use of ppm in this context. Please clarify.

Page 9, lines 282- page10, line 286, This supports my earlier comment that every instrument needs to be evaluated in the in-situ setup employed.

Page 11, line 337-338, This is fairly deep into the manuscript before stating that the calibration factor for ammonia is used for dimethylamine and pyridine. It should be made clear to the reader earlier.

Page 27, Table 1, It should be noted here that only ammonia was directly calibrated and the other calibration factors were assumed or parameterized from other measurements.

Page 29, Figure 1, This figure needs better labeling of the parts, consistent with the description in the text. Include the lengths of tubing. Consider a blow-up insert of the ion source with more detail.

Page 31, Figure 3, Why no x-axis error bars in panel b? Page 9, line 259 seems to suggest that there is a factor of 2 uncertainty in the iodic acid mixing ratio determined by the nitrate CI-APi-TOF.

Page 35, Figure 7. Figure 7 is very busy. The agreement between the water cluster CI-APi-TOF and the PICARRO is mediocre, especially when noting that the ammonia mixing ratio access is logarithmic, though for the most part they are trending in the same direction. However, what is causing the deviation observed 29.10 – 30.10 where the water cluster CI-APi-TOF shows a significant ammonia drop and then increase that does not correlate with fan cycling or temperature changes?

---

## Author Comment (AC1) · 31 Jan 2020

Response to Anonymous Referee 1 on "Measurement of ammonia, amines and iodine species using protonated water cluster chemical ionization mass spectrometry"

We thank the referee for the constructive comments that help improving our manuscript. In the following, the comments of the referee are shown in black, shaded font. Our replies are shown in blue font. Text that has been added or revised in the manuscript is shown in red font.

There could be better clarification throughout the manuscript to indicate that the amine and iodine species are detected and that their mixing ratios are estimated semiquantitatively, as formal calibrations have not been performed with a primary standard.

We agree with the referee that the manuscript should clearly indicate that the mixing ratios for amines and iodic acid are estimated and prone to higher uncertainty compared to the ammonia measurement. We have updated the manuscript according to the specific comments from the referee. Our revised text is listed after our replies.

The title should be revised to reflect quantitation of ammonia and detection of DMA and iodine oxide species specifically instead of 'amines and iodine species' which is misleading. The discussion clearly states that the DMA and HIO3 quantities are estimates based off of scaled numbers from H3O+ or NO3- CI-APi-TOF responses to other species (e.g. NH3 or H2SO4). Converting the units from ncps to mixing ratios for those scaled responses leads to values that may be highly inaccurate. The potential for quantification and sensitive detection of these species by H3O+ CI-APi-TOF can be discussed, but the scaled response estimates should not be used to report mixing ratios, such as in Figures 5 and 8. These should be replaced with signal reported as normalized counts per second (ncps). The authors state in several locations in the discussion potential sources of bias from the assumptions made in converting signal to mixing ratios, which are not fully characterized for the CLOUD chamber and could easily represent a factor of 2 or more error. Based on the presented data, mixing ratios for these compounds should be replaced with ncps units throughout the manuscript. The authors make arguments for the validity of some assumptions throughout the manuscript as well, which seem to be undermined by other parts of the discussion. See the technical comments below for instances where this is the case and the discussion should be revisited.

In principle we agree with the referee that it should clearly be stated that the reported mixing ratios for dimethylamine and iodic acid are not based on a direct calibration with these substances. However, in the following we argue that it is nevertheless justified to report mixing ratios in Figures 5 and 8.

A recent publication by Sipilä et al. (2016) reports on measurements of iodic acid with a nitrate CI-APi-TOF (same as the one used in our study for the HIO3 measurements shown in Fig. 5 on the x-axis). No direct calibration for iodic acid was available at that time and it is still not available at present. However, there is good reason to assume that HIO3 is ionized with the same (or at least similar) efficiency as sulfuric acid (for which the nitrate CI-APi-TOFs are directly calibrated). Sulfuric acid is ionized by NO3- at the collision limit because it is a much stronger acid than HNO3. Compared with HNO3 iodic acid has a much stronger acidity. Furthermore, HIO3 is detected at m/z 175, whereas the strongest signal for sulfuric acid is found at m/z 160 (HNO3.HSO4- cluster), therefore both compounds are found at very similar masses in the mass spectrum and mass discrimination effects of the CI-APi-TOF should be very small. Both arguments indicate that HIO3 can be quantified with the nitrate CI-APi-TOF technique. Therefore, we think it is well-justified to use the concentrations from the nitrate CI-APi-TOF as the reference in Figure 5. However, we have added remarks to the abstract, Table 1 and to the caption of the Figures 3 and 5 that indicate that the reported values are not based on measurements with an instrument that was directly calibrated with HIO3. Furthermore, a comment was added to the Figure 3 caption that the estimated systematic uncertainty for the HIO3 values is +100%/-50% (same as reported by Sipilä et al., 2016).

Regarding the measurement of amines, we have also added remarks to the abstract, the main text, Table 1 and Figure 8, which indicate that no direct calibration was performed for DMA. Furthermore, we replaced "amines" with "dimethylamine" throughout the text and only mention in the outlook that other amines (and diamines) can be measured with the protonated water cluster CI-APi-TOF technique. Nevertheless, we think that the reported mixing ratios for DMA should be kept. DMA has a very high proton affinity (929.5 kJ mol-1), i.e., significantly higher than ammonia (853.6 kJ mol-1). Therefore, DMA should efficiently be ionized by protonated water clusters. Hanson et al. (2011) report that for their water cluster mass spectrometer both ammonia and dimethylamine are ionized with a similar efficiency. This was also reported much earlier by Sunner et al. (1988) although they did not report on the sensitivity of DMA but on other amines with a similar proton affinity. The second argument that supports our assumption that DMA can be quantified is based on the data shown in Figure 8 (Figure 9 in our revisited manuscript). The CLOUD chamber offers in principle the possibility to serve as a calibration system itself. With the known flow rates of DMA into the chamber, the chamber volume and the DMA lifetime (known from wall loss decay experiments) the DMA mixing ratio inside the chamber can be calculated (see, e.g., Simon et al., 2016). Figure 8b) (Figure 9a in our revisited manuscipt) indicates good agreement between estimated DMA (using the sensitivity from the ammonia calibration) and the calculated DMA inside the CLOUD chamber. Taking this together, plus the added caveats about the missing direct calibration (that has now been added to the manuscript) we think that the reported DMA mixing ratios should be kept.

Applying these arguments, also the title can be left unchanged. Especially when the lacking direct calibrations for DMA and HIO3 are immediately mentioned in the abstract below the title.

As said above, we have updated our text and Table 1 to clarify our arguments and to make clear that the only direct calibration was performed with ammonia. Specific changes made in the text and in the Table / Figures are listed after the corresponding comments below.

Intercomparison with the Picarro NH3 cavity ring down instrument does not evaluate the measurements correctly and ignores discussion regarding the limitations of the CI-APi-TOF approach (e.g. response times on the order of hours) or the experimental setup of the intercomparison (e.g. the Picarro connected to the exhaust of the CI-APi-TOF).

Our initial text may have been misleading here. The CI-APi-TOF doesn't have a response time in the order of hours. We just took averages over a time of hours to calculate e.g. the detection limits. The detection limits shown here wouldn't change dramatically if we splitted this 150minute period into e.g. 7 periods of 20 minutes:

| Period   | Calculated detection limit (pptv) |  |
|----------|-----------------------------------|--|
| Period 1 | 0.42                              |  |
| Period 2 | 0.81                              |  |
| Period 3 | 0.39                              |  |
| Period 4 | 0.51                              |  |
| Period 5 | 0.74                              |  |
| Period 6 | 0.39                              |  |
| Period 7 | 0.42                              |  |

The reason for taking a 150 minute instead of a 20 minute period was to get more data points and thus a better statistic to calculate the standard deviation. The initial text at table 1 ("The calculated detection limits are based on a 2.5 hour measurement at 278 K and 80% RH (averaging time of single data points: 1 minute).") may be misleading. Thus, we removed the extra information about the time scale of the measurement. Next to this, we updated our paper by a Figure showing the response time of the CI-APi-TOF between two steady states, which is shown on the following page (Section 3.3 "Response times").

Figure 7 (Figure 8 in our revised manuscript) shows an ammonia decay over several days at the CLOUD chamber. This may be misleading and interpreted as a response time of the insrument. The CLOUD chamber is a 26.1m3 vessel of stainless steel. Once, the chamber walls are saturated with ammonia, it takes hours to remove it (as also shown in the PICARRO trace). This time is not due to instrumental response times. We would like to refer to previous publications, eg Kupc et al. (2011) or Duplissy et al. (2016) that describe the CLOUD chamber in a more detailed way. We updated our text accordingly and we also added limitations of the CI-APi-TOF approach (depletion of primary ions at higher concentrations).

We agree with the referee that an intercomparison as shown in Figure 7 (Figure 8 in our revised manuscript) of our paper would be problematic provided the PICARRO would have been connected to the exhaust of the CI-Api-TOF. However, this was not the case during measurements carried out at the CLOUD chamber. Our previous text may have been misleading in this regard and has been updated accordingly (changes are shown at the specific technical comments). The PICARRO was only connected to the exhaust once (24 hour period) for an intercomparison between both instruments. The setup at the CLOUD chamber is shown in the Figure below.

**3.3 Response times**

The response time of the water cluster CI-APi-TOF is defined as the characteristic time needed for the instrument to react on changes in the ammonia mixing ratio. The response time takes into account two processes. These are the time needed until the instrument reacts on changes in the mixing ratio and the time needed until a steady state is established in the lines. In the following, we define the response time as the time required for the instrument to reach 95% of the new mixing ratio being injected. Figure 4 indicates the typical response times of the water cluster CI-APi-TOF during calibrations (here at 60% relative humidity). It shows a decay between two calibration steps when the injected ammonia is reduced from 9509 pptv to 6911 pptv and a rise in the signal when the ammonia mixing ratio is increased from 500 pptv to 9509 pptv. Panel a) indicates a clear difference between the time needed until the instrument reacts on the changes in the mixing ratio (red line) and the time needed until the lines reach 95% of the new steady state (black line). We expect the same behavior for a decay from 9509 to 500 pptv, however, the mixing ratios were gradually reduced during calibrations. Thus, for the gradual decays, the time needed for the lines to reach a new equilibrium is rather short. While the variation of instrumental response time is small (6 to 10 seconds for decays from 9509 to 6911 pptv and 18 to 25 seconds for a rise from 500 pptv to 9509 pptv, respectively), the time until a steady state is established in the lines varies depending on precursor conditions and relative humidity (see Section 3.8). Thus, an estimation of a response time can vary significantly. In our experiments, the response times (including both processes described above) during a rise in ammonia mixing ratio varied between 535 seconds (20% relative humidity) and 890 seconds (60% relative humidity, shown in Figure 4). For a decay of ammonia mixing ratio from 9509 to 6911 pptv the response times vary between 37 seconds and 54 seconds.

**Figure 4:** Response time of the water cluster CI-APi-TOF during calibrations at 60% RH. The injected ammonia level is shown by the blue line. The signal of the water cluster CI-APi-TOF is shown by the grey line (here the data are shown with a 1 second time resolution (no averaging applied)). The black line shows the response time until a steady state (panel a)) or 95% of the final measured concentration is reached (panel b)). This response time is defined as the sum of the response time of the water-cluster CI-APi-TOF (red line) and the time required until the lines reach a new steady state. See text for details.

**Added in Section 3.7:**

The time from 25.10 to 26.10 shows a steep increase in the PICARRO trace, while the ammonia trace derived from the water cluster CI-APi-TOF flattens out at 20 ppbv of ammonia. This indicates that the primary ions of the water cluster CI-APi-TOF are depleted at high vapor concentrations. It is important to mention that not only ammonia concentrations were elevated at this time, but also other vapor concentrations were rather high. During the CLOUD13 campaign, where a revised version of the ion source was used (see Section 3.2), the significant depletion of primary ions has been observed only at ammonia mixing ratios of 40 ppbv.

For example, there is a clear background offset in the Picarro measurements from the chamber that is not accounted for, but for which the CI-APi-TOF dataset is corrected accordingly through its independent calibration. There are other important measurement concepts that are missed in the data analysis here such as the aforementioned background offset correction, inlet and instrument surface sorption/ desorption effects impact on response times, and detection limits. See the technical comments corresponding to these sections of the manuscript below for several specific comments. It is possible that the intercomparison is technically invalid based on the experimental setup and should be removed from the manuscript, but this cannot be assessed without further data provided by the authors regarding the setup of flows to direct the chamber air to the instruments.

We agree with the referee that a correction for background is important when showing analyzed data. However, the data shown for the CI-APi-TOF are not corrected for a background either (this was done by purpose to show the low instrumental background of the CI-Api-TOF). We corrected the background of the PICARRO in a revised Figure shown below (background of water cluster CI-APi-TOF still not corrected). The PICARRO data shown in Figure 7a (Figure 8a in our revised manuscript) were smoothed using a moving average of 5 minutes in our initial Figure. We updated the Figure using the same 1-minute average as for the CI-APi-TOF here to show that the PICARRO trace is below its detection limit. We used the moving average as the PICARRO trace covers half of the Figure when the 1-minute-average is applied. We updated our Figure, where we now use the Figure shown above that includes the background substraction of the PICARRO.

---

## Author Comment (AC2) · 31 Jan 2020

**Response to Anonymous Referee 2 on "Measurement of ammonia, amines and iodine species using protonated water cluster chemical ionization mass spectrometry"**

We thank the referee for the constructive comments that help improving our manuscript. In the following, the comments of the referee are shown in black, shaded font. Our replies are shown in blue font. Text that has been added or revised in the manuscript is shown in red font.

Though powerful, CIMS is not an absolute measurement technique. A good, defensible calibration is necessary. The manuscript should do a better and clearer job indicating that only the ammonia detection is calibrated with a primary standard and that the mixing ratios for the other species are estimated qualitatively. It is too easy for the reader to lose sight of this fact, since no differentiation between calibrated and estimated mixing ratio results is made in the table or the figures.

We agree with the referee that the manuscript should clearly indicate that the mixing ratios for amines and iodine species are not determined by a direct calibration. We updated the relevant text passages accordingly throughout the manuscript. Major changes are listed below:

- Estimated mixing ratios shown in this manuscript are now marked in tables and figures with an asterisk and a comment that indicates that these mixing ratios are estimated.
- We added a discussion related to our approach of deriving mixing ratios for dimethylamine from the ammonia calibration factor in Section 3.9. We refer to this discussion when mixing ratios of dimethylamine are mentioned.
- We added a comment to Figure 3 stating that the estimated uncertainty for the HIO3 values is +100%/-50% (same as reported by Sipilä et al., 2016).

**Updated changes in the manuscript are detailed in the replies to the technical comments below.**

The main product of this work is the development of the ion source. However, details of the ion source are lacking in the text and figures. Figure 1 is more of a cartoon than a schematic. The details that are given in the text (page 5, lines 143 – page 6, 162) are difficult to translate to Figure 1. For example, a counter electrode and capillary are described in the text but not identified in the figure. Dimensions are given in the text that are not shown in the figure. This makes it unnecessarily difficult for the reader to follow how the ion source truly works and evaluate its performance. Also, details such as tubing length and flow rates for the calibration dilution components should also be given.

We thank the referee for this helpful comment. We added a panel b) to our Figure 1 showing a more detailed drawing of the ion source. This drawing is now also described in our text. The added text is shown in red font below.

**Figure 1:** The experimental setup of the water cluster CI-APi-TOF during ammonia calibration is shown in panel a) The blue color indicates the sample flow. It consists of a mixture of 80% nitrogen and 20% oxygen. A portion of the sample flow can be humidified with a water bubbler ( $H_2O$  aq) to achieve different relative humidities.  $B_1$  represents the ammonia gas bottle, while  $B_2$  represents a gas bottle containing pure nitrogen. There are five mass flow controllers (MFCs; labeled as  $M_{1-5}$ ) allowing two dilution steps. Three MFCs ( $M_1$ ,  $M_2$ ,  $M_3$ ) control the amount of ammonia that is added through a 1/16''capillary into the center of the sample flow, where the second dilution stage occurs. The reagent ions (i.e., protonated water clusters) are produced when the ion source gas (argon, oxygen, water vapor) passes a corona needle at a positive high voltage (detailed in panel b). The calibration setup is disconnected during the measurements at the CLOUD chamber to reduce backgrounds (leakage from the 1/16'' capillary). Details of the ion source used during CLOUD13 are shown in panel b. The primary ions are guided towards the sample flow using a counter electrode (Electrode 1). Additionally, a funnel is used to accelerate the primary ions towards the sample flow. A second electrode (Electrode 2) is installed directly in front of the pinhole of the mass spectrometer. The ions enter the mass spectrometer through a capillary on the top of Electrode 2.

**Modified in Section 2.2 (Line 144):**

A schematic drawing of the calibration setup and the ion source is shown in Figure 1. The gas mixture for the ion source is composed of argon, oxygen and water vapor. It is introduced from two lines placed in the opposite direction to each other at an overall flow rate of ~2.6 slm (Figure 1a). The Electrodes of the ion source are displayed in red colors in Figure 1b. The connection to the mass spectrometer is shown using blue color. The 1" sampling line and the inlet (22 mm inner diameter) consist of stainless steel and are shown in green color. Components used for insulation are shown in white colors. A total sample flow rate of ~ 19.5 slm is maintained by a vacuum pump and a mass flow

controller. The overall length of the sampling line connecting the CLOUD chamber and the ion molecule reaction zone is 1.3 m. A voltage of 3600 V is applied to the corona needle while 500 V are applied to the conically-shaped counter electrode (Electrode 1 in Figure 1b) made of stainless steel. The housing of the ion source is made of polyether ether ketone (PEEK). The ion source gas and the generated reagent ions flow through a funnel (smallest inner diameter 2.5 mm) before they mix with the sample flow. A small capillary (inner diameter of 0.8 mm) is located opposite of the funnel (Electrode 2 in Figure 1b). The electric field between the counter electrode and the capillary (at ground potential) accelerates the ions towards the entrance of the mass spectrometer. The pinhole plate (pinhole inner diameter of 350  $\mu$ m) and the capillary are in electric contact and ~0.8 slm flow through the capillary and the pinhole into the mass spectrometer. The measured product ions are generated in the ion-molecule reaction zone (IMR, yellow area in Figure 1a) at atmospheric pressure. The dimension of the IMR is defined by the distance between the counter Electrode and the capillary (~ 16.4 mm). After passing the pinhole, the ions are transported through two quadrupoles (Small Segmented Quadrupole, SSQ and Big Segmented Quadrupole, BSQ) towards the detection region of the mass spectrometer (Micro-Channel Plate, MCP; pressure is approx.  $1 \times 10^{-6}$  hPa). The estimated reaction time is

---

## Author Response (AR2)

**Response to Anonymous Referee 1 on "Measurement of ammonia, amines and iodine species using protonated water cluster chemical ionization mass spectrometry"**

We thank the referee for the constructive comments that help improving our manuscript. In the following, the comments of the referee are shown in black, shaded font. Our replies are shown in blue font. Text that has been added or revised in the manuscript is shown in red font. In addition to the changes proposed by the reviewer, we have also added linguistic changes to the manuscript that do not change the content. These are listed after the answers to the reviewer, whereby we list the sentence previously used first.

*Lines 306-307: The Authors should be clear that the data is not shown in the Figure. Where the response times corresponding to the Figure are discussed (Lines 309-312), it may be easier to relate between the two if the response times are reported in minutes (consistent with the x-axis in the figure).*

We agree with the referee that a clearer distinction between the response times discussed in Section 3.3 is helpful to improve understanding this Section of the manuscript. Thus, we changed the text in our Section 3.3 to distinguish in a more clear way between the response times shown in the Figure and the response times reported in the text. Next to this, we changed the unit of our x label in the Figure towards "seconds" to stay consistent with the text.

We define the response time as the time required for the instrument to reach 95% of the new mixing ratio being injected. The response time takes into account two processes. It includes both the response time of the instrument ("instrumental response time") and the time for the lines to reach a steady state for ammonia delivery ("line response time"). Figure 4 indicates the typical response times of the water cluster CI-APi-TOF during calibrations (here at 60% relative humidity). It shows a decay between two calibration steps when the injected ammonia is reduced from 9509 pptv to 6911 pptv and a rise in the signal when the ammonia mixing ratio is increased from 500 pptv to 9509 pptv. Panel a) indicates a clear difference between the fast instrumental response time (red line) and the slower line response time (black line). While we expect a similar behavior in instrumental response time for a decay from 9509 to 500 pptv, a longer line response time is expected due to re-evaporation of ammonia from the sampling lines. Thus, the mixing ratios were gradually reduced during calibrations. The instrumental response time shown in Figure 4 is at 6 s for a decay in mixing ratio (9509 pptv to 6911 pptv) and at 18 s for a rise (500 pptv to 9509 pptv). The line response time is at 37 s (decay) and at 890 s (rise). The difference between the short line response time for a decay in ammonia mixing ratio compared to a rise  The experiments were repeated several times at varying relative humidies.

The instrumental response time only varied by a few seconds during our experiments (between 6 to 10 s (decay) and 18 to 25s (rise)). While the variation in instrumental response time is small, the line response time can vary strongly depending on precursor conditions and relative humidity. During our experiments, the line response time varied between 37 s and 54 s (decay) and between 535 s and 890 s (rise) respectivelly. Interactions of ammonia with the sampling line are discussed in Section 3.8.

[Figure]

**Figure 4:** Response time of the water cluster CI-APi-TOF during calibrations at 60% RH. The injected ammonia mixing ratio from MFC settings is shown by the blue line. The signal of the water cluster CI-APi-TOF is shown by the orange line (here the data are shown with a 1 second time resolution i.e. no time-averaging is applied). The green line represents 95% of the mixing ratio being applied with the next MFC setting. The black line shows the response time until a steady state (panel a) or 95% of the final measured concentration is reached (panel b). The response time is the sum of the response time of the water-cluster CI-APi-TOF (red line) and the (slower) response time for the lines to reach a steady state where the walls are conditioned.

*Line 373: There is a citation formatting issue here for von Bobutzki et al, with the 'von' missing.*

We updated the text according to the comment of the referee. Next to this, we replaced "in any case" by "nevertheless":

Nevertheless, the detection limit derived for ammonia is well below the LOD reported for other measurement techniques and instruments (von Bobrutzki et al., 2010; You et al., 2014; Lin; Wang et al., 2015).

*Section 3.7: The discussion here is, generally, quite well presented. One detail remains, however, regarding the comparison of the PICARRO against the CI-APi-TOF due to reporting NH3 measurements from the PICARRO well below its detection limit (stated as 366 pptv on Line 237). At Line 419, the 'background' for the PICARRO is reported as ~200 pptv, but this is lower than its detection limit. This should be replaced with a statement clearly stating that the backgrounds were below the detection limit. This same approach is necessary in discussing the detection of NH3 by the PICARRO in the subsequent comparison. Where PICARRO NH3 is less than 366 pptv, the discussion should not comment on the performance of the instrument as it cannot detect such levels (e.g. 'can hardly detect this decrease in the VMR' on Lines 428-429 and again in the last sentence of the section at Lines 449-450). Likewise, NH3 data from the PICARRO that is below the LOD should either not be presented in Figures 8a and 8b or a horizontal line on the plot denoting the PICARRO LOD should be provided so readers are not expecting more from this instrument.*

We changed our Section 3.7 according to the comments from the Referee. The detection limit stated as 366 pptv was determined during the CLOUD13 campaign (during the same time where we determined the detection limit of the water cluster CI-APi-TOF). In Section 3.7, we show a longterm timeseries from the CLOUD12 campaign (one year before). We determined the detection limit at this time to be at 170.1 pptv. On Line 231 (Section 2.4) we added the detection limit during the CLOUD12 experiments to the description of the PICARRO:

By using the same method (at the same time period) as for the water cluster CI-APi-TOF (see Section 3.5), we derive a detection limit of 366.2 pptv for the PICARRO unit used in our study during CLOUD13. The detection limit derived for the PICARRO used during the CLOUD12 experiments is at 170.1 pptv.

Next to this, we updated the text in Section 3.7 on Line 408 of our updated manuscript, where we initially stated:

"Figure 8a shows the measurements of the water cluster CI-APi-TOF, the PICARRO, and the calculated value for ammonia. The signal measured by the water cluster CI-APi-TOF follows the injected ammonia almost instantaneously (first injection is on Oct. 23), whereas the PICARRO only shows elevated concentrations above its background of ~200 pptv much later."

The updated text is shown below:

Figure 8a shows the measurements of the water cluster CI-APi-TOF, the PICARRO and the calculated value for ammonia. The PICARRO trace is shown for the time when the detection limit (170.1 pptv during CLOUD12) is exceeded. The signal measured by the water cluster CI-APi-TOF follows the injected ammonia almost instantaneously (first injection is on Oct. 23).

The initial aim behind commenting on the PICARRO measurements below its detection limit was to explain deviations between the instruments at low mixing ratios of ammonia. Nevertheless, we agree with the Referee that stating the detection limit of the PICARRO in the beginning of the Section already explains deviations between the instruments at low concentrations. Thus, we removed the sentence on line 428-429 (previous manuscript): "Due to the higher LOD, the PICARRO, however, can hardly detect this decrease in the VMR."

In order to bring the following sentence back into context with the text ("*This increased response time can be explained by a combination of the longer sampling line (~1.8 m compared to 1.3 m for the water cluster CI-APi-TOF), the lower flow rate (~ 1 slm with a core sampling of 5 slm compared with ~ 20 slm for the water cluster CI-APi-TOF) and the higher detection limit of the PICARRO.*"), we have changed it slightly:

The slower response time of the PICARRO can be explained by a combination of the longer sampling line (~1.8 m compared to 1.3 m for the water cluster CI-APi-TOF) and the lower flow rate (~ 1 slm with a core sampling of 5 slm compared with ~ 20 slm for the water cluster CI-APi-TOF).

In the last sentence of the Section, we now explain why the PICARRO trace is not shown in Figure 8b (since the mixing ratios are below the LOD of the PICARRO):

Due to its higher detection limit, the PICARRO is insensitive at these low mixing ratios (green line in Figure 8b).

We updated Figure 8 according to the comments of the Referee, where we removed the PICARRO trace below its detection limit and where we added a line stating the detection limit of the PICARRO in Figure 8b instead:

[Figure]

*Figure 6: The temperature color scale can be removed as only two temperatures are on this figure. The corresponding T values can be added to the caption and/or the legend for each of the three data series.*

We agree with the Referee that adding the temperature to the caption and removing the color scale improves legibility since only two temperatures are shown. Thus, we updated our Figure 6 as shown below:

[Figure]

*Figure 7: Should the 'primary ions' here be 'reagent ions' in order to be consistent with the figure caption?*

Using 'reagent ions' in Figure 7 instead of 'primary ions' makes sense to stay consistent with our caption and text. Thus, we updated Figure 7:

[Figure]

Next to the changes suggested by the Referee, we tried to improve the legibility of our manuscript. The content has not been changed. Since all changes are minor text corrections, we don't comment on the changes made. The original sentences from our previous manuscript are shown in shaded, blue font, the updated sentences are shown in shaded, red font.

Since we use "compounds" instead of "species" throughout the manuscript, we would like to change the title from:

*"Measurement of ammonia, amines and iodine species using protonated water cluster chemical ionization time of flight mass spectrometry".*

Towards:

*"Measurement of ammonia, amines and iodine compounds using protonated water cluster chemical ionization time of flight mass spectrometry".*

We changed our Figure 1, where we adjusted Panel b) so, that the ion source has the same orientation as shown in Panel a. Next to this, we changed the color of the arrows pointing towards the ion source to improve the clarity of the figure. We added an elipse around the ion source in Panel a and an arrow to connect the two panels with each other:

[Figure]

Line 19:

*[…] at the CLOUD (Cosmics Leaving OUtdoor Droplets) […].*

*[…] at the CERN CLOUD (Cosmics Leaving OUtdoor Droplets) […].*

Line 21-23:

[revised manuscript text omitted]

Line 209-210:

*[…]a calibration factor for HIO$_3$ is derived […],  that is calibrated for sulfuric acid […]*

*[…]a calibration factor for HIO$_3$ has been derived […] , which itself had been calibrated for sulfuric acid […]*

Line 212:

*However, as the reaction of sulfuric acid with nitrate ions is at the kinetic limit, the detection limits […]*

*However, as the reaction of sulfuric acid with nitrate ions is at the kinetic limit (Viggiano et al., 1997), the detection limits […]*

Line 214-215:

*[…]Thus, the applied assumption in the present study was also used in a previous study for deriving gas phase concentrations of iodic acid (Sipilä et al., 2016).*

*[…]The assumption we use in the present study was also applied in a previous study for deriving gas phase concentrations of iodic acid (Sipilä et al., 2016).*

Line 217:

*[…] experimental runs were chosen […]*

*[…] experimental runs were selected […]*

Line 220-221:

*A PICARRO G1103-t NH$_3$ Analyzer (PICARRO Inc., USA) measured ammonia mixing ratios based on based on cavity-ring down spectroscopy during CLOUD12 and CLOUD13.*

*A PICARRO G1103-t NH$_3$ Analyzer (PICARRO Inc., USA) measuring ammonia mixing ratios based on cavity-ring down spectroscopy was also connected to the CLOUD chamber during CLOUD12 and CLOUD13.*

Line 226:

*[…]to minimize line losses and to decrease the response times.*

**[…]to minimize line losses and to shorten the response time.**

Line 230:

*[…]a lower detection limit of 200 pptv is reported […]*

**[…]a lower detection limit of 200 pptv is specified […]**

Line 236:

*$(H_2O)H_3O^+$ is the dominant primary ion in the mass spectrum.*

**The dominant primary ion is $(H_2O)H_3O^+$.**

Line 238-239:

*[…]the signal from $NH_4^+$ can have a similar magnitude compared with the signal from $H_2O^+$ (possibly from reactions of $O_2^+$ and $H_2O$).*

**[…]the signal from $NH_4^+$ has a similar magnitude as $H_2O^+$ (which may arise from reactions of $O_2^+$ with $H_2O$).**

Line 240:

*[…]is essential in terms of reaching low detection limits […]*

**[…]is essential to reach low detection limits for ammonia […]**

Line 241:

*At the low masses the APi-TOF used in the present study reaches a resolving power of ~2000 Th/Th […]*

**At low mass the APi-TOF used in the present study has a resolving power of ~2000 Th/Th […]**

Line 244:

*Prominent peaks from $N_2H^+$, $NO^+$ and $O_2^+$ can also be found in the spectrum shown in Figure 2.*

**Prominent peaks from $N_2H^+$, $NO^+$ and $O_2^+$ are also seen (Figure 2).**

Line 246:

*[…] as reagent ions (in equation (1)) as no evidence exists that they interact […]*

*[…] as reagent ions (in equation 1) as we have no indication that they interact […]*

Line 248:

*[…]; but $NH_3^+$ is not considered […]*

*[…]; however $NH_3^+$ is not considered […]*

Line 252-253:

*For estimating an ammonia mixing ratio according to equation (1), the product ion count rates are normalized against the dominating reagent ion count rates.*

*For estimating an ammonia mixing ratio (Equation 1), the signal ion count rates are normalized to the dominating reagent ion count rates.*

Line 253-254:

*[…]the reagent ion signals are significantly higher than the product ion count rates.*

*[…] the reagent ion signals are an order of magnitude higher than the signal ion count rates.*

Line 254-255:

*This indicates that no significant reagent ion depletion occurs and thus the normalized counts per second respond linearly with the ammonia VMR […]*

*This indicates that very little reagent ion depletion occurs and thus the normalized counts per second are linear with the ammonia mixing ratio […]*

Line 257:

*Figure 3 shows the calibration curves obtained for $NH_3$ and $HIO_3$ (for the CLOUD13 campaign).*

*Figure 3 shows the calibration curves obtained for $NH_3$ and $HIO_3$ during the CLOUD13 campaign.*

Line 259-260:

*(ammonia: $NH_4^+$ and $(H_2O)NH_4^+$; iodic acid: $HIO_3H^+$ and $HIO_3H_3O^+$).*

*($NH_4^+$ and $(H_2O)NH_4^+$ for ammonia and $HIO_3H^+$ and $HIO_3H_3O^+$ for iodic acid).*

Line 260-261:

*[…]VMR (on the x-axis)[…] and the uncertainty of the VMR inside the ammonia gas bottle.*

*[…]mixing ratio (x-axis) […] and the uncertainty of the ammonia gas bottle concentration.*

Line 269-271:

[revised manuscript text omitted]

Line 502-503:

*The technique has unprecedented low detection limits regarding the ammonia measurement as well as a fast time response and time resolution.*

**The instrument has demonstrated unrivalled low detection limits for ammonia, as well as a fast time response and time resolution.**

Line 503-505:

*A next step is its application to atmospheric measurements. The technique should be suitable for such measurements as the amount of clean gas required (ca. 2 slm of argon and some oxygen) is rather small and can easily be supplied with gas bottles (one argon gas bottle, 50 liters at 200 bar should last ~3 days).*

**We believe this instrument can readily be applied to atmospheric measurement. The amount of clean gas required for the corona tip (2 slm of argon and some oxygen) is rather small and can easily be supplied with gas bottles (one argon gas bottle of 50 liters at 200 bar should last about 3 days).**

Line 506:

*[…]can be taken from the chamber.*

**[…]can be drawn from the chamber.**

Line 510-511:

*[…]further signals corresponding to $NH_4^+$, $NDH_3+$, $ND_2H_2^+$ were also visible in the spectra due to deuterium-hydrogen exchange, which makes this method unfavorable.*

**[…]further signals corresponding to $NH_4^+$, $NDH_3+$, $ND_2H_2^+$ were also present due to deuterium-hydrogen exchange, which makes this method unsuitable.**

Line 511-512:

*[…] is also unfavorable since […] even with a high resolution mass spectrometer.*

**[…] is also unsuitable since the […] even for a high resolution mass spectrometer.**

Line 514:

*[…] which leads to greatly reduced sampling line losses and improved time response during ammonia measurements.*

*[…] which reduces sampling line losses and sharpens the time response for ammonia measurements.*

Line 516-517:

*We also tested the passivation technique, however, the […]*

*We tested this passivation technique but found it unsuitable for our instrument; the […]*

Line 518:

*[…]led to a consumption of the reagent ions since […]*

*[…] led to excessive consumption of the reagent ions since […]*

Line 523-525:

*The set-up and characterization of a water cluster Chemical Ionization-Atmospheric Pressure interface-Time Of Flight mass spectrometer (CI-APi-TOF) is described.*

*We have described the design and performance of a novel water cluster Chemical Ionization-Atmospheric Pressure interface-Time Of Flight mass spectrometer (CI-APi-TOF) for measurements of ammonia, amines (dimethylamine) and iodine compounds.*

Line 525-526:

*The generated protonated water clusters ($(H_2O)_{n\geq1}H_3O^+$) are used to selectively ionize compounds […]*

*The protonated water clusters ($(H_2O)_{n\geq1}H_3O^+$) selectively ionize compounds […]*

Line 528-529:

*The water cluster CI-APi-TOF was used at the CLOUD chamber where very low background ammonia mixing ratios (ca. 4 pptv at 278 K) were achieved.*

*The water cluster CI-APi-TOF was operated at the CLOUD chamber where very low background ammonia mixing ratios were measured (4 pptv at 278 K).*

Line 529-530:

*The level of detection (LOD) was estimated as 0.5 pptv for $NH_3$. To our knowledge, such a low detection limit for ammonia measurements has not been reported so far.*

*The limit of detection (LOD) for ammonia was estimated as 0.5 pptv. To our knowledge, such a low detection limit for ammonia measurements is unprecedented.*

Line 532:

*[…] was observed when using nitrogen instead of argon.*

*[…]was observed when nitrogen was used instead of argon.*

Line 532-534:

*Although, the sensitivity towards the measurement of NH$_3$ depends somewhat on the relative humidity of the sample flow, the observed sensitivity changes were rather low and can be taken into account by a correction factor.*

*The instrument shows some sensitivity to relative humidity of the sample flow (factor 3 increase in signal from 5% to 80% RH). However, this can be readily measured and corrected.*

Line 537:

*[…] iodine species. A total of 29 different iodine-containing compounds were unambiguously identified […]*

*[…] iodine compounds. A total of 29 different iodine-containing compounds were identified […]*

Line 539:

*[…] measuring iodic acid during CLOUD.*

*[…] measuring iodic acid.*

Line 541:

*[…] for which the nitrate CI-APi-TOF is calibrated for.*

*[…] for which the nitrate CI-APi-TOF was calibrated for.*

Line 541-542:

*The estimated LOD for the water cluster CI-APi-TOF regarding iodic acid was as low as 0.007 pptv.*

*In this way, we estimated the LOD for iodic acid in the water cluster CI-APi-TOF to be 0.007 pptv.*

Line 544-545: We added the following sentence:

*Laborarory and ambient measurements indicate increased importance of ammonia for new particle formation and growth in both pristine and polluted environments.*

Line 547-548:

*Airborne measurements in the upper troposphere, where very low ammonia mixing ratios can be expected (Höpfner et al., 2016) should in principle be feasible as well. For such measurement the water cluster CI-APi-TOF technique should be very well-suited due to the very low LODs that can be realized.*

*The water cluster CI-APi-TOF technique is also well-suited for airborne measurements in the upper troposphere, where fast response times and low detection limits are vital (Höpfner et al., 2016).*

List of relevant changes made in the manuscript

- We revised our Section 3.3 ("response times"). We updated the xlabel of our Figure 4 towards seconds (s) in order to be consistent with the text, where we give response times in seconds. Next to this, we changed our description in Section 3.3, where we now distinguish between "instrumental response time" and "line response time" in a more clear way. We also state, which response times are shown in our figure 4.
- We revised our Section 3.7 ("CLOUD chamber characterization"), where we now state the detection limit of the PICARRO in the beginning of the discussion. Next to this, the discussion related to response times at low ammonia concentrations has been updated accordingly.
- We made minor corrections to our figures 1, 6 and 7 to improve clarity and to stay more consitent with our text.
- We have made several minor text corrections throughout the manuscript to improve legibility.
- To stay consistent with the text, where we mainly use "iodine compounds", we changed the title of our manuscript from "*
[revised manuscript text omitted]

960